J Physiol 603.17 (2025) pp 4867–4886

4867

# Sex-specific effects of betamethasone on glucocorticoid and apoptotic signalling pathways in the sheep placenta

Ashley S. Meakin[1] , Mitchell C. Lock[1] , Stacey L. Holman[1] , Joshua L. Robinson[2] ,
Vicki L. Clifton[3] , Claire T. Roberts[4] , Michael D. Wiese[5], Kathryn L. Gatford[2]
and Janna L. Morrison[1]

[1]Early Origins of Adult Health Research Group, Clinical and Health Sciences, University of South Australia, Adelaide, SA, Australia
[2]Robinson Research Institute and School of Biomedicine, University of Adelaide, Adelaide, SA, Australia
[3]Mater Medical Research Institute, University of Queensland, Brisbane, QLD, Australia
[4]Flinders Health and Medical Research Institute, College of Medicine and Public Health, Flinders University, Adelaide, SA, Australia
[5]Centre for Pharmaceutical Innovation, Clinical and Health Sciences, University of South Australia, Adelaide, SA, Australia

Handling Editors: Laura Bennet & Rebecca Simmons

The peer review history is available in the Supporting information section of this article (https://doi.org/10.1113/JP289044#support-information-section).

The Journal of Physiology

**Abstract figure legend** This study investigated placental molecular and structural responses to betamethasone treatment in the near-term sheep pregnancy. We found that, 48 h after betamethasone treatment, female, but not male, placentae had higher expression of the pro-apoptotic glucocorticoid receptor (GR) isoform, GRαC. This molecular adaptation coincided with higher expression of apoptotic and oxidative stress markers, lower endoplasmic reticulum (ER) stress markers and an increase in placental barrier thickness compared to control females. Collectively, our findings indicate a heighted apoptotic response and altered glucocorticoid signalling in females in response to betamethasone may contribute to a greater risk of betamethasone-induced placental insufficiency.

A. S. Meakin, M. C. Lock, K. L. Gatford and J. L. Morrison contributed equally to this study.

**Abstract** Antenatal corticosteroid therapy (ACS, e.g. betamethasone) is standard clinical care for pregnancies at risk of preterm delivery to reduce the incidence of neonatal lung disease and death. Variable and sex-specific impacts of ACS on the placenta have been reported and may reflect differing expression profiles of glucocorticoid receptor (GR) isoforms. We therefore examined placental GR isoforms and molecular and structural responses to betamethasone in the clinically relevant sheep pregnancy. Pregnant Merino ewes at 138 days of gestation (term = 150 days) received I.M. injections of saline or 11.6 mg of betamethasone 48 and 24 h prior to Caesarean section delivery of lambs and tissue collection. Placental glucocorticoid concentrations were measured using liquid chromatography-tandem mass spectrometry. Markers of GR signalling and placental development and function were measured using histology, western blotting and quantitative real-time PCR. Betamethasone increased diffusion barrier thickness in female placentae only and reduced placental cortisol concentrations in both sexes. Betamethasone increased cytoplasmic GR$\alpha$C, GR-P and GR$\alpha$D isoforms in female placentae only; neither treatment nor sex impacted nuclear GR isoform expression. Expression of angiogenic genes was higher, whereas that of growth-promoting genes was lower, in betamethasone-exposed placentae, independent of sex. Similarly, expression of endoplasmic reticulum stress genes was lower in betamethasone-exposed than control placentae, whereas those involved in oxidative stress and apoptosis were higher, particularly in females. Betamethasone induced molecular changes in the placenta within 48 h of exposure. The apoptotic response was heightened in female placentae, possibly driven by higher expression of specific GR isoforms, which contributes to a greater risk of ACS-induced placental insufficiency.

(Received 9 April 2025; accepted after revision 16 July 2025; first published online 10 August 2025)
**Corresponding authors** A. S. Meakin and J. L. Morrison: Early Origins of Adult Health Research Group, Health and Biomedical Innovation, Clinical and Health Sciences, University of South Australia, GPO Box 2471, Adelaide, SA 5001, Australia. Email: ashley.meakin@unisa.edu.au and Janna.Morrison@unisa.edu.au

## Key points

- Betamethasone treatment for pregnancies at risk of preterm delivery not only reduces the risk of neonatal death, but also acts on glucocorticoid receptors (GR) in the placenta, inducing sex-specific changes that may impact function and fetal growth. In this study, we explored sex-specific placental molecular responses to betamethasone in the clinically relevant sheep pregnancy.
- Betamethasone increased markers of angiogenesis and decreased markers of growth and proliferation in placentae of both sexes.
- In females only, betamethasone increased expression of the pro-apoptotic GR isoform, GR$\alpha$C, which coincided with an enriched pro-apoptotic response and an increased placental diffusion barrier thickness, indicative of placental insufficiency.
- Our findings highlight that betamethasone induces several molecular changes in the sheep placenta within 48 h of exposure and supports previous sex-specific findings in other species, indicative of a conserved female response.

## Introduction

The placenta is a highly vascular, transient organ that is critical for maternal and fetal health, as well as pregnancy success. It not only supports fetal growth and development by exchanging nutrients and waste between maternal and fetal circulations, but also regulates the fetal and maternal endocrine and immune systems, protects the fetus from excess endogenous and exogenous chemical exposure, and is critical for the regulation of fetal programming and thus later life offspring health (Burton et al., 2016). It is therefore essential that normal placental function is maintained throughout pregnancy to ensure optimal fetal wellbeing. However, changes to the maternal–placental–fetal glucocorticoid milieu can impact placental function, impair fetal growth and increase the risk of intrauterine morbidity and mortality (Meakin et al., 2021). For example, excess maternal

concentrations of the glucocorticoid, cortisol, can activate the glucocorticoid receptor (GR) and disrupt organ function, including that of the placenta (Clifton, 2010). Aberrant GR-mediated signalling in the fetoplacental unit can lead to fetal growth restriction (FGR) and an increased risk of neurodevelopmental perturbations and cardiometabolic diseases in later life (Morrison, 2008; Morrison et al., 2012).

Despite the known risks associated with excess endogenous glucocorticoids, maternally administered antenatal corticosteroid therapy (ACS; dexamethasone or betamethasone) is standard clinical care for pregnancies at risk of preterm delivery. Unlike cortisol, ACS passes from maternal to fetal circulations without placental inactivation and thus interacts with GR in the fetal lungs to promote lung maturation (Jobe et al., 2003; Liggins, 1969; Morrison et al., 2012; Seckl, 1997) and reduce neonatal lung disease and death (McGoldrick et al., 2020). However, as the GR is expressed ubiquitously (Lockett et al., 2024), ACS also activates the GR in other fetal organs and in the placenta (Cuffe et al., 2017; Ninan et al., 2022; Räikkönen et al., 2020).

Within human and rodent placenta, ACS inhibits trophoblast growth and proliferation, reduces the abundance of placenta-specific growth-related factors, and impairs placental growth (Audette et al., 2010; Arias et al., 2021). ACS also alters the transcriptome of mouse placenta, downregulating pathways involved in growth and proliferation and upregulating pathways involved in apoptosis (Baisden et al., 2007). Additionally, ACS induces changes to the placental methylome that indicate placental dysfunction and inflammation (Czamara et al., 2021). *In vitro*, ACS inhibits vascular endothelial growth factor (VEGF)-induced tube formation of human umbilical vascular endothelial cells in a GR-dependent manner (Logie et al., 2010). In baboon placentae, betamethasone reduced indices of endothelial nitric oxide synthase (eNOS) function (Aida et al., 2004). By contrast, other studies report that ACS improves placental function including decreased placental vascular resistance (Wallace & Baker, 1999) and accelerated placental villus maturation (Um-Bergström et al., 2018) in humans, and increased

umbilical artery diameter and blood flow in mice (Cahill et al., 2019). These inconsistencies between studies may reflect the timing and duration of ACS exposure, or may be a result of differences in the expression profile of placental GR isoforms at the time of ACS administration (Lockett et al., 2024). Indeed, changes in GR isoform subcellular localisation and abundance can modulate tissue- and cell-specific responses to endogenous and exogenous corticosteroids (Whirledge & DeFranco, 2018). Moreover, placenta-specific glucocorticoid signalling pathways differ between sexes (Clifton et al., 2019; Cuffe et al., 2017; Saif et al., 2014, 2015, 2021), such that glucocorticoid sensitivity is greater in females, potentially conferring a survival adaptation in instances of intra-uterine perturbations. However, whether ACS impacts placental molecular function in a sex-dependent manner is largely unknown.

The expression profile of GR isoforms in the sheep placenta is similar to that of human placenta (Clifton et al., 2019; Saif et al., 2014). Because human female placentae are more glucocorticoid responsive than male placentae (Clifton, 2010; Saif et al., 2014), we hypothesised that betamethasone exposure would induce sex-specific differences to molecular and structural measures in the near-term sheep placenta, with a greater response observed in females than males.

## Methods

### Ethical approval

The South Australian Health and Medical Research Institute (SAHMRI) animal ethics committee approved this study (SAM455.19) according to the Australian code for the care and use of animals for scientific purposes (National Health & Medical Research Council, 2013). All investigators adhered to the ethical principles outlined by Grundy (2015), the principles of the 3Rs: Replacement, Reduction and Refinement (Tannenbaum & Bennett, 2015), and the ARRIVE guidelines (Percie du Sert et al., 2020).

**Ashley S. Meakin** is an Australian National Heart Foundation Postdoctoral Fellow in the Early Origins of Adult Health Research Group, University of South Australia. His research focuses on two main areas: (1) understanding and targeting androgen signalling in pregnancy to improve generational cardiometabolic outcomes and (2) exploring the impact of pregnancy complications on drug and steroid metabolism in the maternal, placental and fetal compartments. Through this research, Meakin hopes to improve pregnancy outcomes to ensure that every baby gets the most out of life. **Mitchell C. Lock** Postdoctoral Fellow in the Early Origins of Adult Health Research Group, University of South Australia. His research focuses on understanding the molecular mechanisms by which suboptimal *in utero* conditions, such as fetal hypoxaemia, can influence lung maturation and cardiac development and the programming of adult cardio-respiratory disease. Through this research, Mitchell hopes to improve pregnancy outcomes to ensure that every baby gets the most out of life.

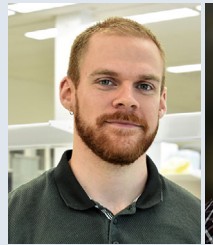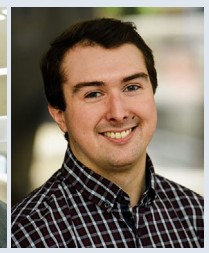

## Animal model

The snimals used in the present study were from our previously reported cohort of an experimental model of mild allergic maternal asthma (Hammond et al., 2025; Robinson et al., 2024; Roff et al., 2025). Animals were group-housed in outdoor paddocks with *ad libitum* access to water and natural pasture supplemented with oaten hay as required. To investigate the impact of betamethasone on placental molecular responses, singleton and twin-bearing asthmatic ewes were randomised using a random number generator to receive saline or betamethasone (Celestone Chronodose, 11.6 mg; Schering Plough, Baulkham Hills, NSW, Australia), with I.M. injections 48 and 24 h prior to Caesarean section (Robinson et al., 2024), consistent with clinical practice (Antenatal Corticosteroid Clinical Practice Guidelines Panel, 2015). A dose of medroxyprogesterone acetate of 150 mg was given I.M. to all ewes 6–9 days prior to antenatal injections to prevent premature labour from betamethasone-induced progesterone withdrawal in ewes receiving treatment (Jenkin et al., 1985; Jobe et al., 2003).

## Caesarean sections and tissue collection

At $140 \pm 2$ days of gestational age (dGA), fasted ewes were anaesthetised I.V. with ketamine (7 mL kg$^{-1}$) and diazepam (0.3 mL kg$^{-1}$), then intubated, and anaesthesia was maintained with isoflurane (1.5–2.5% in air; Lyppards, Beverley, SA, Australia). Fetuses were delivered by Caesarean section and ventilated for 45 min as previously reported (Robinson et al., 2024). Upon completion of lung function studies, ewes and lambs were humanely killed with sodium pentobarbitone (20 mg kg$^{-1}$; Virbac Australia, Peakhurst, NSW, Australia). Major maternal and neonatal organs were sampled as previously described (Clifton et al., 2016; McBride et al., 2021; Meakin et al., 2022) and stored for future studies, including phenotyping and sampling (Vatnick et al., 1991; Zhang et al., 2016) of placentomes from each pregnancy (saline: female, $n = 6$, male $n = 8$; betamethasone: female, $n = 7$, male, $n = 5$). Cross-sections were consistently cut at the centre of each placentome to minimise variability between pregnancies. All morphometric and molecular analyses were performed on type B placentomes.

## Placental morphometric measures

Sagittal sections (5 µm) from type B placentomes (saline; female, $n = 5$, male $n = 8$; betamethasone; female, $n = 7$, male, $n = 4$) were stained with Masson's trichrome, using published methods (Clifton et al., 2019; Fletcher et al., 2007; MacLaughlin et al., 2005; Roberts et al., 2001; Ward et al., 2006; Zhang et al., 2016). One section per animal was scanned using a NanoZoomer-XR (Hamamatsu,

Hamamatsu, Japan). Ten random-systematic sampled fields of view at 20× magnification were captured 1 mm apart by an individual blinded to treatment group and fetal sex. The proportions of placental trophoblast, fetal capillaries, fetal connective tissue, maternal epithelium, maternal capillaries and maternal connective tissue were quantified, using point counting with an isotropic L-36 Merz transparent grid overlayed onto each counting frame, and counting 360 points in each section. The number of points necessary to achieve a standard error of less than 10% was calculated from a preliminary study using a nomogram relating test point number and volume density (Weibel et al., 1966). The volume density (Vd) of each of the specified components of the placentome and placenta was calculated, using the following formula, $Vd = Pa/PT$, where Pa is the total number of points falling on a particular component and PT is the total number of points in the section (Fletcher et al., 2007; MacLaughlin et al., 2005; Roberts et al., 2001; Zhang et al., 2016). The estimated volume of each of the specified components in each placentome was calculated by multiplying the volume density of each component by the weight of the individual placentome.

Intercept counting using the same grid and fields of view was utilised to calculate the surface density (Sv) of trophoblast, taking into account the magnification, using the formula, $Sv = 2 \times Ia/LT$, where Ia is the number of intercepts with the L36 Merz line and LT is the total length of the lines applied. The total surface area for the placentome from which the section was cut was estimated by multiplying the surface density by placentome weight. The arithmetic mean barrier thickness of maternal epithelium/feto-maternal syncytium was calculated using the formula, barrier thickness, $B_T = VdME/S_V$, where VdME is the volume density of maternal epithelium and Sv is the surface density of trophoblast.

## Quantification of placental apoptotic cells

Type B placentomes (saline; female, $n = 5$, male, $n = 8$; betamethasone; female, $n = 7$, male, $n = 4$) were sectioned (5 µm) and floated onto SuperFrost Plus slides (VWR International, Radnor, PA, USA). Slides were incubated at 60°C for 1 h followed by deparaffinisation and rehydration. Endogenous peroxidase activity was blocked with 3% hydrogen peroxide (Sigma-Aldrich, St Louis, MO, USA), followed by heat-induced antigen retrieval in citrate buffer (pH 6.0). Slides were incubated overnight with antibodies against cleaved caspase-3 (dilution 1:300; ASP175; Cell Signaling Technology, Danvers, MA, USA), a marker of apoptosis, at 4°C following incubation with non-immune serum (eBioscience IHC/ICC Blocking Buffer; Thermo Fisher Scientific, Waltham, MA, USA) to prevent non-specific binding (Zhang et al., 2024).

Negative control slides with the primary antibody omitted were used to confirm the lack of non-specific binding of the secondary antibody or reagent contamination (Lock et al., 2015; Zhang et al., 2024). Additional negative control slides, where the diluted primary antibody was replaced by rabbit serum (Sigma-Aldrich) at the same protein concentration, were used to confirm the absence of non-specific staining. Negative controls were incubated overnight at 4°C in parallel under the same experimental conditions. Slides were subsequently incubated with secondary antibody [goat anti-rabbit IgG (H+L) secondary antibody, Biotin; Thermo Fisher Scientific] and streptavidin protein-horseradish peroxidase (Thermo Fisher Scientific). All sections were counterstained with Mayer's hematoxylin (Sigma-Aldrich) for visualisation of nuclei. One slide per animal was scanned using a NanoZoomer-XR (Hamamatsu, Hamamatsu, Japan). QuPath, version 0.5.1 (Bankhead et al., 2017) and used to determine the number of cleaved caspase-3 positive cells. Five $500 \times 500$ μm frames were selected by systematic randomised sampling on each slide by a trained user blinded to the treatment groups. An average of $1590 \pm 172$ nuclei was analysed per frame, and the percentage of positive nuclei was calculated as the mean of data from all five frames.

### Quantification of placental protein abundance

Snap-frozen type B placentomes (saline; female, $n = 6$, male, $n = 7$; betamethasone; female, $n = 6$, male, $n = 5$) were used for tissue subcellular fractionation and western blotting, as previously described (Meakin, Nathanielsz, Li, Clifton, et al., 2024). Membranes were blocked in 5% bovine serum albumin in Tris-buffered saline with 1% Tween for 1 h at room temperature, and then incubated overnight with: anti-GR (dilution 1:1000; catalog. no. A303-491°; Bethyl Laboratories, Montgomery, TX, USA); anti-GRβ (dilution 1:500; Cat no. PA3-514; Thermo Fisher Scientific); anti-Bcl-$X_L$ (dilution 1:500; catalog. no. sc-8392; Santa Cruz Biotechnology, Santa Cruz, USA); anti-Bad (dilution 1:500; catalog. no. sc-943; Santa Cruz Biotechnology); or anti-phospho-Bad (Ser136; dilution 1:500; catalog. no. sc-7999; Santa Cruz Biotechnology) antibodies (Amanollahi et al., 2025; Botting et al., 2014; Orgeig et al., 2015). The appropriate secondary antibodies were applied for 1 h. Membranes were subsequently probed with anti-β actin (dilution 1:4000; catalog. no. A300-491A; Bethyl Laboratories) as a loading control. SuperSignal West Pico Chemiluminescent Substrate (Thermo Fisher Scientific) was used to detect reactive bands by enhanced chemiluminescence. Western blots were imaged using ImageQuant LAS 4000 (GE Healthcare, Chicago, IL, USA) and protein abundance was determined by densitometry using Image Quant software (GE Healthcare), with values for each sample normalised to their respective loading control. To allow comparisons of subcellular GR isoform expression between membranes, expression data for each subcellular fraction were further normalised to the expression of matched GR isoforms in pooled samples that were included on each membrane (Meakin, Nathanielsz, Li, Clifton, et al., 2024).

### Quantification of placental mRNA expression

All essential information regarding the quantitative real-time PCR procedure is included as per the MIQE guidelines (Bustin et al., 2009). RNA was extracted from snap-frozen type B placentomes (saline; female, $n = 6$, male, $n = 7$; betamethasone; female, $n = 6$, male, $n = 5$) using QIAzol Lysis Reagent solution and RNeasy purification columns in accordance with the manufacturer's guidelines (Qiagen, Hilden, Germany). Total RNA was quantified by spectrophotometric measurement on a NanoDrop Lite (Thermo Fisher Scientific) and RNA integrity was assessed using agarose gel electrophoresis. cDNA was synthesised from 1 μg of total RNA using the Superscript III First Strand Synthesis System (Invitrogen, Waltham, MA, USA) in accordance with the manufacturer's guidelines. Controls containing either no RNA transcript or no Superscript III were used to test for reagent contamination and genomic DNA contamination, respectively. The geNorm component of qbaseplus, version 3.4 (Biogazelle, Ghent, Belgium) was used to determine the most stable reference genes from a panel of candidate reference genes and the minimum number of reference genes required to calculate a stable normalisation factor, as previously described (Lie et al., 2014; McGillick et al., 2013; Soo et al., 2012). Three stable reference genes [tyrosine 3-monooxygenase/tryptophan 5-monooxygenase activation protein zeta (YWHAZ), TATA-box binding protein (TBP) and beta actin (ACTB)] were run in parallel with all target genes using Qiagen QuantiNova SYBR Green (Qiagen) on a QuantStudio 7 Pro Real-time PCR system (Applied Biosystems, Foster City, CA, USA). Target genes were chosen *a priori* to investigate key pathways involved in placental function (Table 1). The abundance of each transcript relative to the abundance of stable reference genes was calculated and expressed as mRNA mean normalised expression (MNE).

### Quantification of placental glucocorticoids

Hormone concentrations in snap-frozen type B placentomes (saline; female, $n = 6$, male, $n = 7$; betamethasone; female, $n = 6$, male, $n = 5$) were determined by liquid chromatography (LC) (Shimadzu Nexera XR, Shimadzu, Japan) coupled to a SCIEX

**Table 1. Primer details.**

| Gene | Forward primer | Reverse primer |
| --- | --- | --- |
| *ACTB* | CCAAGGCCAACCGTGAGA | AGCCTGGATGGCCACGT |
| *ANGPT1* | TTGCCATAACCAGTCAGAG | AACCACCAGCCTCCTGTTA |
| *ANGPT2* | AGAACCAGACCGCTGTGATG | TGCAGTTTGCTTATTTCACTGGT |
| *ATF6* | AACCACGCAGCTACCTAATC | CTGTCTCCTTAGCACAGCAATA |
| *BAX* | CAGGATGCATCCACCAAGAAGC | TTGAAGTTGCCGTCGGAAAACAT |
| *BCL2* | GTGGAGGAGCTCTTCAGGGA | GTTGACGCTCTCCACACACA |
| *BNIP3* | GTTCCCGACTCTGCTTCTATTT | GTCACAGTGGGAGCTCTTG |
| *CAT* | GTCACAGTGGGAGCTCTTG | CGCCTTGGAGTATCTGGTAATG |
| *EIF2AK3* | CCTTCGGAAGCTTCTCCTTATG | CCGGAGCGCAGTTAGTTTAT |
| *ERN1* | CAGGAGTACGTGGAACAGAAG | GGCATGGAGAGGAGGATATTG |
| *FGF* | AGAACGGAAGCTCCAAACTC | AGCACGGCCAATGGTAAA |
| *GPX* | GTGGCACCATCTATGAGTACG | CACGTTGACGAAGAGGATGTAT |
| *HSD11B1* | GCGCCAGATCCCTGTCTGAT | AGCGGGATACCACCTTCTTT |
| *HSD11B2* | GAGACATGCCGTTTCCATGC | TGATGCTGACCTTGACACCC |
| *IGF1* | TTGGTGGATGCTCTCCAGTTC | AGCAGCACTCATCCACGATTC |
| *IGF1R* | AAGAACCATGCCTGCAGAAGG | GGATTCTCAGGTTCTGGCCATT |
| *IGF2* | GCTTCTTGCCTTCTTGGCCTT | TCGGTTTATGCGGCTGGAT |
| *IGF2R* | GATGAAGGAGGCTGCAAGGAT | CCTGATGCCTGTAGTCCAGCTT |
| *IL1B* | TGCCTACGAACATGTCTTCCGTGA | TGCTCTCTGTCCTGGAGTTTGCAT |
| *IL6* | TCATCCTGAGAAGCCTTGAGA | TTTCTGACCAGAGGAGGGAAT |
| *IL8* | CAGGATTCACGAGTTCCTGTT | CTTCCACATGTCCTCACATCTC |
| *KI67* | TCAGTGAGCAGGAGGCAGTA | GGAAATCCAGGTGACTTGCT |
| *NOX4* | GGCAAGAGAACAGACCTGATTA | CACCGAGGACGTCCAATAAA |
| *NR3C1* | ACTGCCCCAAGTGAAAACAGA | ATGAACAGAAATGGCAGACATTTTATT |
| *P53* | GCTATGGGTCGACTCGCCGC | GGGGACTGCGCCTCACAACC |
| *PCNA* | ACTCCACTGTCTCCTACAGTAA | CGATCTTGGGAGCCAAATAGT |
| *PGF* | GGCACCTGCCCTCTATTTATTA | GGGTCTGTCTTCTTTCTCTCAC |
| *SOD1* | CTTCGAGGCAAAGGGAGATAAA | ACTGGTACAGCCTTGTGTATTG |
| *SOD2* | AGTAAACCGTCAGCCTTACAC | CCACGCTCAGAAACACTACA |
| *TBP* | GCTGAGGAAGTCGCCAAGAA | GCATAAGGTGGGAGGCTGTT |
| *VEGF* | TGTAATGACGAAAGTCTGGAG | TCACCGCCTCGGCTTGTCACA |
| *VEGFR1* | CCGAAGGGAAGAAGGTGGTC | GACTGTTGTCTCGCAGGTCA |
| *VEGFR2* | TTGATTGCTGGCATGGGGAT | AGGCAGAGAGAGTCCCGAAT |
| *YWHAZ* | TGTAGGAGCCCGTAGGTCATCT | TTCTCTCTGTATTCTCGAGCCATCT |

6500 Triple-Quad system (tandem mass spectrometry; SCIEX, Framingham, MA, USA) as previously described (McBride et al., 2021; Meakin, Nathanielsz, Li, Huber, et al., 2024). Prepared samples were injected onto an ACQUITY UPLC BEH C18 Column 130Å, 1.7 μm, 2.1 × 100 mm (Waters Corp, Milford, MA, USA). Mobile phases were 0.2% formic acid in water (A) and 0.2% formic acid in acetonitrile (B). Flow rate was 0.3 mL min$^{-1}$ and mobile phase B was initially 10% and increased linearly to 90% over 10 min and then held at 90% for 2 min, after which it returned to 10% over 3 min prior to injection of the next sample. Hormone concentrations were calculated via integration with a standard curve that ranged from 0.05 to 100 ng mL$^{-1}$.

**Statistical analysis**

Statistical analysis was performed using SPSS, version 28 (IBM Corp., Armonk, NY, USA). Up to one potential outlier was detected using Grubbs' test and omitted prior to subsequent analyses. Data were analysed by linear mixed model using treatment (Saline or BETA) and fetoplacental sex as factors; ewe was included as a random factor to correct for the effects of the maternal environment. Where interactions were significant, Bonferroni-adjusted pairwise comparisons were performed. Data are presented as the mean ± SD, unless stated otherwise. $P < 0.05$ was considered statistically significant.

**Table 2. Cohort fetal and placental characteristics.**

|  | Saline | | BETA | | $P_{treatment}$ | $P_{sex}$ | $P_{intx}$ |
|---|---|---|---|---|---|---|---|
|  | Female (n = 6) | Male (n = 8) | Female (n = 7) | Male (n = 5) |  |  |  |
| Body weight (kg) | 4.55 ± 0.54 | 4.74 ± 0.69 | 4.13 ± 0.75 | 4.53 ± 0.79 | 0.387 | 0.477 | 0.982 |
| Total placenta weight (g) | 574 ± 133 | 528 ± 178 | 471 ± 85 | 475 ± 188 | 0.175 | 0.404 | 0.551 |
| Average weights of placentomes (g) |  |  |  |  |  |  |  |
| Type A | 7.3 ± 4.1 | 8.3 ± 3.5 | 6.9 ± 4.1 | 4.9 ± 4.3 | 0.358 | 0.839 | 0.422 |
| Type B | 7.8 ± 3.7 | 7.9 ± 5.2 | 6.4 ± 3.8 | 6.8 ± 5.4 | 0.714 | 0.721 | 0.886 |
| Type C | 9.9 ± 2.9 | 8.6 ± 4.6 | 8.9 ± 2.6 | 6.6 ± 6.5 | 0.415 | 0.295 | 0.772 |
| Type D | 12.7 ± 7.6 | 11.8 ± 6.2 | 2.2 ± 0.7 | 7.9 ± 8.7 | **0.036** | 0.439 | 0.295 |
| Average proportions of each placentome type within each placenta (%) |  |  |  |  |  |  |  |
| Type A | 39 ± 35 | 36 ± 30 | 34 ± 24 | 23 ± 33 | 0.475 | 0.605 | 0.740 |
| Type B | 41 ± 21 | 24 ± 9 | 43 ± 20 | 49 ± 27 | 0.193 | 0.756 | 0.289 |
| Type C | 19 ± 5 | 27 ± 24 | 34 ± 18 | 21 ± 9 | 0.533 | 0.684 | 0.159 |
| Type D | 16 ± 20 | 22 ± 17 | 33 ± 30 | 42 ± 34 | 0.087 | 0.481 | 0.931 |

Data are presented as the mean ± SD. Statistical analysis: linear mixed model (fixed factors: treatment and sex; random factor: ewe; intx = interaction effect). BETA = betamethasone. $P < 0.05$ was considered statistically significant and is indicated in bold.

## Results

### Cohort characteristics

Neonatal body weight did not differ between treatments or fetal sexes (Table 2). Total placental weight and average weights of type A, B and C placentomes did not differ between treatments or sexes; however, the average weight of type D placentomes was lower in betamethasone-exposed than saline placentae, irrespective of sex (Table 2). The proportions of type A, B, C and D placentomes did not differ between treatments or sexes (Table 2).

### Markers of angiogenesis are higher in placentae exposed to betamethasone

Expression of angiopoietin 2 (*ANGPT2*), placental growth factor (*PGF*), VEGF receptor 1 (*VEGFR1*) and *VEGFR2* was higher in betamethasone-exposed than saline placentae, irrespective of sex (Fig. 1*B* and *D–F*). *VEGF* and *ANGPT1* mRNA expression did not differ between treatments or sexes (Fig. 1*A* and *C*).

### Markers of growth and proliferation are altered in placentae exposed to betamethasone

Expression of insulin-like growth factor (IGF) 2 (*IGF2*) mRNA was higher in betamethasone-exposed than saline placentae, irrespective of sex (Fig. 2*C*). IGF1 receptor (*IGF1R*) and *IGF2R* mRNA expression did not differ between treatments or sexes (Fig. 2*A*, *B* and *D*). Fibroblast growth factor (*FGF*), antigen Kiel 67

(*KI67*) and proliferating cell nuclear antigen (*PCNA*) mRNA expression were lower in betamethasone-exposed placentae, irrespective of sex (Fig. 2*E–G*).

### Placental maternal epithelium barrier thickness was higher in females exposed to betamethasone

The Vd per placentome of maternal epithelium was higher in males than females, irrespective of maternal treatment (Table 3). The Vd per placentome of maternal capillary, maternal connective tissue, trophoblast, fetal capillary and fetal connective tissue, as well as placentome surface density, did not differ between treatments or sexes (Table 3). The mean arithmetic barrier thickness ($B_T$) of maternal epithelium/feto-maternal syncytium, comprising the distance between maternal and fetal circulations, was higher in betamethasone-exposed than saline placentae, but only in females (Table 3).

### Placental cortisol concentrations are lower in betamethasone-exposed placentae

Placental cortisol concentrations were lower after betamethasone exposure in both sexes and were higher in males than females, but only in saline-treated pregnancies (Fig. 3*A*). Cortisone concentrations did not differ between treatments or sexes; however, the cortisol/cortisone ratio was higher in betamethasone-exposed than saline placentae (Fig. 3*A*). Expression of 11-$\beta$ hydroxysteroid dehydrogenase type 1 (*HSD11B1*), *HSD11B2* and nuclear receptor subfamily 3 group C member 1 (*NR3C1*, encodes GR) mRNA did not differ between treatments or sexes (Fig. 3*B*). Expression of the

**Table 3. Placental structure.**

|  | Saline | | BETA | | $P_{treatment}$ | $P_{sex}$ | $P_{intx}$ |
|---|---|---|---|---|---|---|---|
|  | Female (*n* = 6) | Male (*n* = 8) | Female (*n* = 7) | Male (*n* = 4) | | | |
| Maternal epithelium (Vd/placentome) | 1.08 ± 0.18 | 1.94 ± 0.97 | 1.60 ± 0.25 | 1.96 ± 0.48 | 0.415 | **0.046** | 0.276 |
| Maternal capillary (Vd/placentome) | 0.42 ± 0.21 | 0.55 ± 0.30 | 0.51 ± 0.24 | 0.44 ± 0.14 | 0.771 | 0.836 | 0.173 |
| Maternal connective tissue (Vd/placentome) | 2.44 ± 0.48 | 3.08 ± 1.29 | 2.89 ± 0.63 | 2.32 ± 0.07 | 0.623 | 0.939 | 0.073 |
| Trophoblast (Vd/placentome) | 5.18 ± 1.13 | 6.26 ± 2.71 | 5.42 ± 0.54 | 6.95 ± 3.11 | 0.639 | 0.152 | 0.997 |
| Fetal capillary (Vd/placentome) | 0.60 ± 0.32 | 0.54 ± 0.19 | 0.76 ± 0.25 | 0.85 ± 0.33 | 0.054 | 0.914 | 0.532 |
| Fetal connective tissue (Vd/placentome) | 1.47 ± 0.49 | 1.63 ± 0.87 | 1.74 ± 0.35 | 1.64 ± 0.38 | 0.706 | 0.942 | 0.588 |
| Surface density ($S_V$) | 271 ± 67 | 218 ± 31 | 227 ± 70 | 254 ± 96 | 0.896 | 0.652 | 0.163 |
| Barrier thickness ($B_T$) of maternal epithelium (μm) | 4.2 ± 0.5 | 6.4 ± 1.4 | 5.7 ± 1.4* | 5.4 ± 1.2 | 0.093 | 0.672 | **0.013**[#] |

Vd = volume density; placentome = placentome weight. Data are presented as the mean ± SD. Statistical analysis: linear mixed model (fixed factors: treatment and sex; random factor: ewe; intx = interaction effect) with Bonferroni *post hoc* test. $P < 0.05$ was considered statistically significant and is indicated in bold.
* Significantly different compared to the sex-matched saline group.
[#] $B_T$ of maternal epithelium (μm) was higher in female betamethasone-exposed placentae ($P = 0.003$) but not males ($P = 0.596$) compared to the sex-matched saline group.

glucocorticoid-responsive gene interleukin (IL) 1*β* (*IL1B*) was lower in betamethasone-exposed placentae, irrespective of sex, whereas *IL6* and *IL8* expression did not differ between treatments or sexes (Fig. 3*C*).

## Placental GR isoform expression profiles are impacted by betamethasone in a fetal sex-specific manner

Six GR isoforms (GR*α*A, GR*β*, GR*α*C, GR-P, GR-A and GR*α*D) were identified in cytoplasmic and nuclear

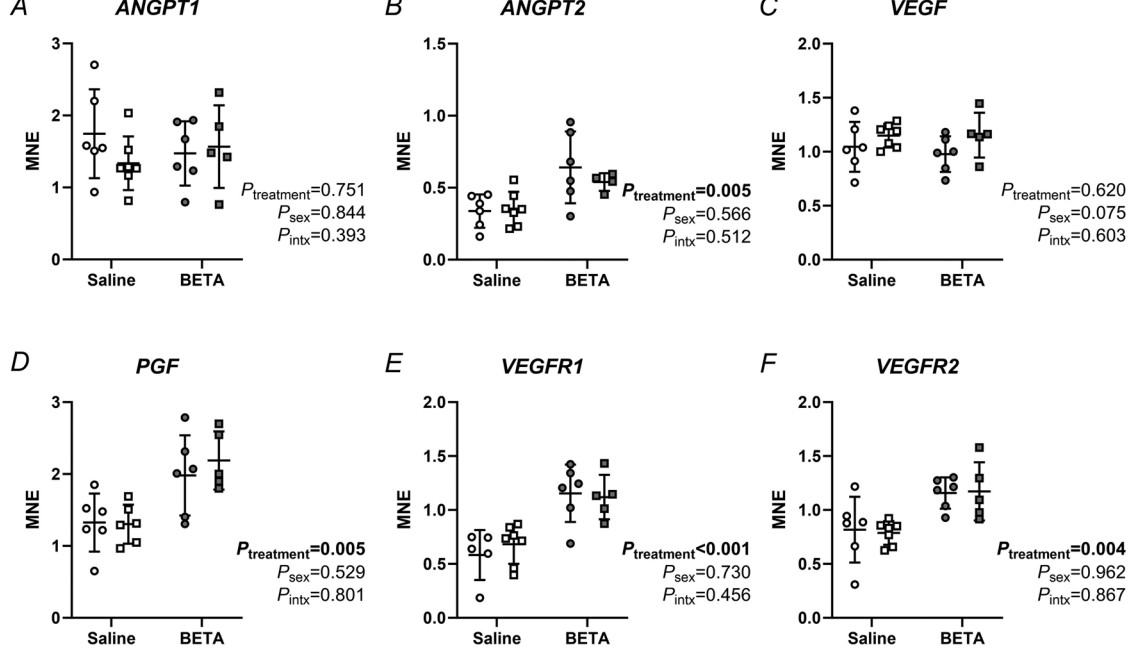

**Figure 1. Placental mRNA expression of angiogenic factors is increased by betamethasone**
Transcript expression of (*A*) *ANGPIT1*, (*B*) *ANGPT2*, (*C*) *VEGF*, (*D*) *PGF*, (*E*) *VEGFR1* and (*F*) *VEGFR2* was measured in female (circles) and male (squares) type B placentome samples from saline (open symbols, female, *n* = 6, male, *n* = 7) and betamethasone (BETA; closed symbols, female, *n* = 6, male, *n* = 5) treatment groups. MNE = mean normalised expression. Statistical analysis: linear mixed model (fixed factors: treatment and sex; random factor: ewe; intx = interaction effect). $P < 0.05$ was considered statistically significant and is indicated in bold. Symbols show data from individual animals, whereas bars and whiskers indicate the mean ± SD.

fractions of sheep placenta (Fig. 4*C*). Additional protein bands that were immunoreactive with the anti-GR (total GR) antibody were identified in both subcellular fractions but have not been confirmed as GR isoforms. Cytoplasmic protein expression of GR$\alpha$C, GR-P and GR$\alpha$D were higher in betamethasone-exposed than saline placentae within females, but not males (Fig. 4*A*). Protein expression of nuclear-localised GR isoforms did not differ between treatments nor sexes (Fig. 4*B*).

### Betamethasone induces fetal sex-specific pro-apoptotic changes in the sheep placenta

Protein expression of the anti-apoptotic factor Bcl-xL was lower in betamethasone-exposed than saline placentae regardless of sex (Fig. 5*A* and *B*). Additionally, mRNA expression of the anti-apoptotic gene *BCL2* (Fig. 5*C*) was lower in betamethasone-exposed than saline placentae and lower in female than male placentae, independent of treatment. Within females only, *BAX* mRNA expression (Fig. 5*C*) and cleaved-caspase 3 staining (Fig. 5*D* and *E*) were higher in betamethasone-exposed placentae (Fig. 5*C*). Protein expression of Bad, phospho-Bad and the phospho-Bad:Bad ratio (Fig. 5*A* and *B*) and mRNA expression of *BNIP3* (Fig. 5*C*), a protein with roles in both apoptosis and autophagy, did not differ between treatments nor sexes.

### Expression of endoplasmic reticulum (ER) stress, oxidative stress and DNA damage-regulating genes are impacted by betamethasone and fetal sex

The mRNA expression of antioxidants *CAT* (Fig. 6*D*) and inner-mitochondria localised *SOD2* (Fig. 6*H*) were both higher in betamethasone-exposed than saline placentae, regardless of sex. mRNA expression of the pro-oxidant *NOX4* (Fig. 6*F*) was also higher in betamethasone-exposed placentae and was higher in female than male placentae. The mRNA expression of *EIF2AK3* (Fig. 6*B*), a marker of ER stress, as well as of DNA damage marker *TP53* (Fig. 6*I*), were lower in betamethasone-exposed placentae, and higher in female than male placentae. Interestingly, mRNA expression of the cytoplasmic antioxidant enzyme *GPX* was lower in betamethasone-exposed placentae, but only in females (Fig. 6*E*). The mRNA expression of ER stress markers *ATF6* (Fig. 6*A*), *ERN1* (Fig. 6*C*) and outer-mitochondrial membrane antioxidant *SOD1* (Fig. 6*G*) did not differ between treatments nor sexes.

### Discussion

In the present study, we have characterised the impact of clinically-relevant betamethasone treatment and fetoplacental sex on placental molecular determinants of

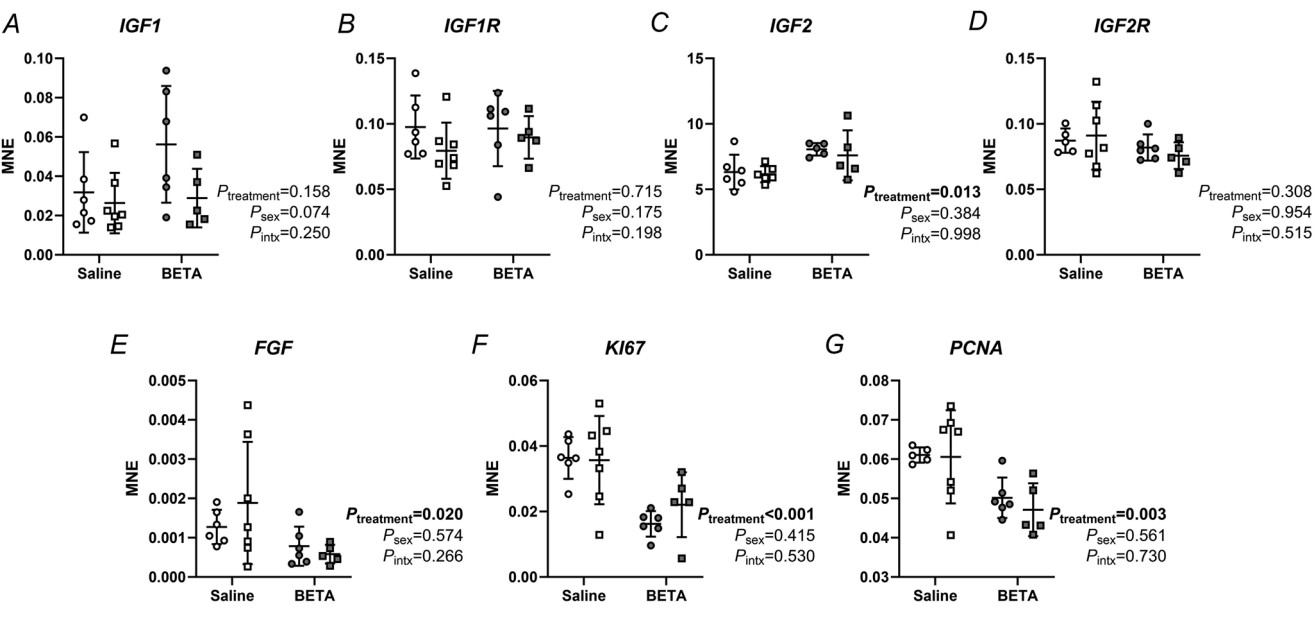

**Figure 2. Placental expression of genes involved in growth and proliferation are impacted by betamethasone**
Transcript expression of (*A*) *IGF1*, (*B*) *IGF1R*, (*C*) *IGF2*, (*D*) *IGF2R*, (*E*) *FGF*, (*F*) *KI67* and (*G*) *PCNA* was measured in female (circles) and male (squares) placenta samples from saline (open symbols, female, $n = 6$, male, $n = 7$) and betamethasone (BETA; closed symbols, female, $n = 6$, male, $n = 5$) treatment groups. MNE = mean normalised expression. Statistical analysis: linear mixed model (fixed factors: treatment and sex; random factor: ewe; intx = interaction effect). $P < 0.05$ was considered statistically significant and is indicated in bold. Symbols show data from individual animals, whereas bars and whiskers indicate the mean ± SD.

function in the pregnant sheep. Our work has identified changes in signalling pathways involved in growth, angiogenesis, proliferation and apoptosis in placentae exposed to betamethasone compared to placentae from saline-treated ewes. The differences in GR isoform expression in the present study are consistent with those previously reported in rats exposed to dexamethasone (Cuffe et al., 2017), where expression of the pro-apoptotic GR isoform, GRα-C, was higher in placentae exposed to ACS, but only in females. This conserved, female-specific expression pattern may contribute to the heightened apoptotic signature that we observed after betamethasone-exposure in female placentae. Collectively, our findings demonstrate

substantive impacts of betamethasone on molecular signatures in the sheep placenta 48 h after maternal betamethasone administration and provide further evidence for sex-specific GR-mediated responses to the intrauterine environment.

Our study identified no impact of a clinically relevant dosage regimen of betamethasone given near-term on lamb body weight, a finding that is consistent with research in humans, sheep and non-human primate studies (Blanco et al., 2014; Elfayomy & Almasry, 2014; Fletcher et al., 2002; Moss et al., 2001; Shields et al., 2012). By contrast, repeat courses of ACS treatment can reduce birthweight outcomes (Moss et al., 2001; Murphy et al., 2008; Wapner et al., 2006), which may be a result

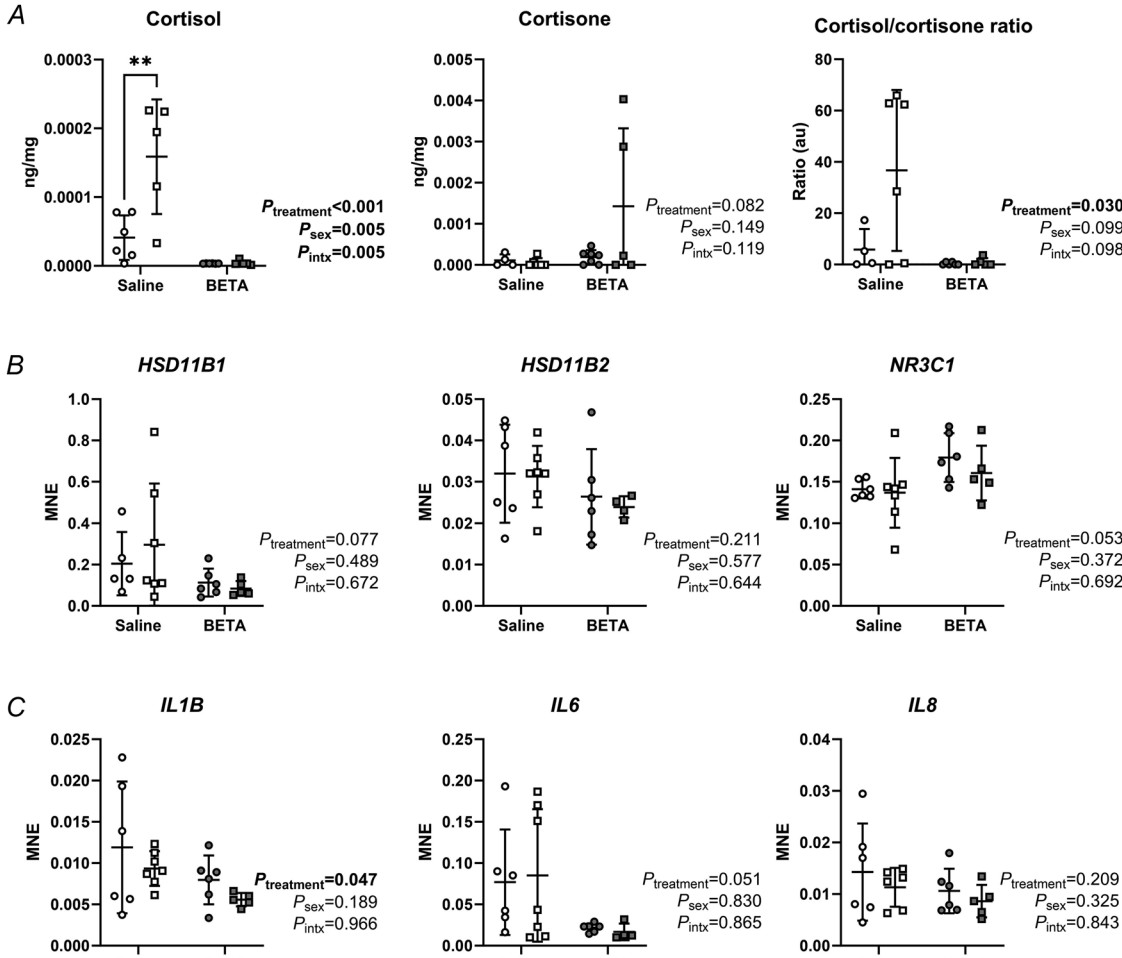

**Figure 3. Placental glucocorticoid concentrations and expression of glucocorticoid responsive genes are impacted by betamethasone exposure**

*A*, tissue-specific concentrations of cortisol, cortisone and the cortisol/cortisone ratio; transcript expression of (*B*) glucocorticoid signalling and metabolism (*HSD11B1*, *HSD11B2* and *NR3C1*); and (*C*) glucocorticoid-responsive inflammatory cytokines (*L1B*, *IL6* and *IL8*) were measured in female (circles) and male (squares) type B placentomes from saline (open symbols, female, $n = 6$, male, $n = 7$) and betamethasone (BETA, closed symbols; female, $n = 6$, male, $n = 5$) treatment groups. MNE = mean normalised expression. Statistical analysis: linear mixed model (fixed factors: treatment and sex; random factor: ewe; intx = interaction effect) with Bonferroni *post hoc* test. $P < 0.05$ was considered statistically significant and is indicated in bold. ***post hoc* $P < 0.01$. Symbols show data from individual animals, whereas bars and whiskers indicate the mean $\pm$ SD.

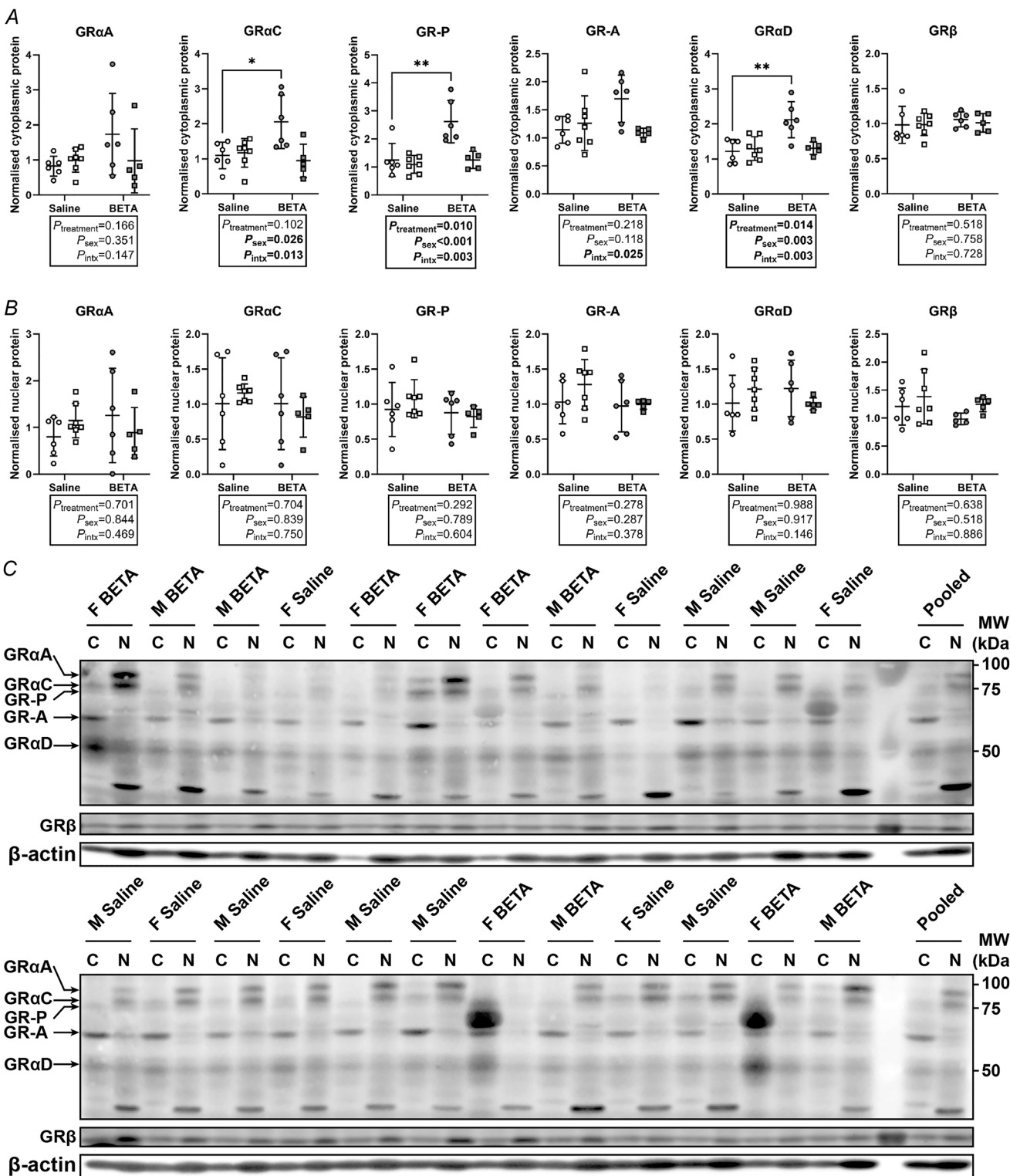

**Figure 4. Expression of glucocorticoid receptor (GR) isoforms is impacted by betamethasone in female sheep placentae only**

*A*, cytoplasmic *c* and (*B*) nuclear (N) expression of GR isoforms of female (circles) and male (squares) type B placentome samples from saline (open symbols; female, *n* = 6, male, *n* = 7) and betamethasone (BETA, closed symbols; female, *n* = 6, male, *n* = 5) treatment groups. *C*, western blots of C and N protein extracts of female and male placenta samples from saline and BETA treatment groups. Six known GR isoforms were detected with molecular weights ranging from 50 to 95 kDa. Membranes were probed with *β*-actin as a loading control. Statistical

analysis: linear mixed model (fixed factors: treatment and sex; random factor: ewe; intx = interaction effect) with Bonferroni *post hoc* test. $P < 0.05$ was considered statistically significant and is shown in bold. *post hoc* $P < 0.05$; **post hoc* $P < 0.01$. Symbols show data from individual animals, whereas bars and whiskers indicate the mean $\pm$ SD.

of chronic ACS-induced placental growth perturbations and dysfunction. Although we observed no effect of betamethasone exposure on placental weight, we did observe lower placental expression of *KI67* and *PCNA*, both markers of cell proliferation and thus organ growth. Other studies exploring the fetoplacental response to ACS have reported placental growth impairments that drive FGR outcomes (Alawadhi et al., 2022; Namdar Ahmadabad et al., 2016; Trifunović et al., 2022). If this reduced growth pattern persisted after betamethasone exposure, we would expect to see reductions in placental size in pregnancies that continue until term, which is of concern because ~40% of ACS-treated pregnancies clinically continue to term (Ninan et al., 2023), with reduced placental size and function potentially compromising fetal growth and well-being. Thus, the timing of tissue collection and duration of ACS exposure in the present study may explain the lack of change in placental and fetal growth.

Placental vascular development, maintenance and appropriate adaptation throughout pregnancy are critical for the continued supply of substrates from maternal to fetal circulations to support fetal growth. Both endogenous and exogenous glucocorticoids exhibit potent vasoactive properties in a cell-, tissue- and species-specific manner (Morgan et al., 2018). However, there are inconsistencies regarding the impact of ACS exposure and placental vasculature development and function. Some studies have identified that ACS reduces placental expression of angiogenic factors in rats and humans (Hewitt et al., 2006; Logie et al., 2010; Ozmen et al., 2015, 2017) and decreases placental eNOS expression in baboons (Aida et al., 2004), culminating in impaired placental function and reduced fetal growth. By contrast, within the context of preterm delivery, ACS administration decreased placental vascular resistance (Wallace & Baker, 1999) and was associated with placental morphological changes indicative of enhanced oxygen and nutrient transfer (Um-Bergström et al., 2018). ACS treatment in a rodent model of hypoxic pregnancy decreased placental vascular resistance, measured as increased umbilical artery diameter, blood flow and reflection coefficient by ultrasound, but only in females (Cahill et al., 2021). More recently, betamethasone improved placental function in a rat model of experimental chorioamnionitis (Netsanet et al., 2025).

In the present cohort, we found higher placental expression of *ANGPT2*, *PGF*, *VEGR1* and *VEGR2*, which are important regulators of placental angiogenesis and vasculogenesis, after betamethasone treatment. This response may facilitate placental vascular remodelling leading to increased oxygen and nutrient exchange from maternal to fetal circulations, thereby compensating for betamethasone-induced lower placental cell division, and resulting in similar neonatal body weights between treatment groups. Considering that ACS induces either pro- or anti-angiogenic responses in the placenta, a response that may be mediated by the timing and duration of ACS administration, the upregulation of pro-angiogenic factors in the present study collectively suggests a molecular adaptation that favours placental vascular function 48 h after betamethasone treatment.

Despite the higher expression of angiogenic factors in betamethasone-exposed placentae, we found only one change in placental structure in response to betamethasone treatment. Barrier thickness was higher after betamethasone-exposure, but only in females. Similar thickening of the placental exchange barrier has been reported in a pregnant guinea-pig model of undernutrition-induced FGR (Roberts et al., 2001) and in human pregnancies complicated by FGR (Macara et al., 1996; Teasdale & Jean-Jacques, 1988). This increase in placental barrier thickness may result in decreased oxygen and nutrients exchange, which may indicate placental insufficiency. Indeed, it would be interesting in future studies to assess the impact of betamethasone treatment on placental function, structure and molecular changes beyond 48 h and determine whether placental changes impact sex-specific impacts of betamethasone on fetal growth.

Our finding that betamethasone reduces placenta-specific cortisol concentrations is consistent with our previous finding in the same cohort of lower neonatal plasma and lung cortisol concentrations in betamethasone-exposed near-term lambs (Robinson et al., 2024), a response that is a result of the negative feedback regulation of glucocorticoid release in the perinatal period (Ballard et al., 1975, 1980; Kajantie et al., 2004; Parker et al., 1996). Interestingly, we also found higher placental cortisol concentrations in males than females, but only in saline-exposed pregnancies, which may indicate sex differences to the bidirectional interconversion of cortisol and cortisone. Despite previous reports that betamethasone increases $11\beta$HSD2 expression (Ma et al., 2003; Ni et al., 2018), we found no difference in gene expression between treatments and sexes; however, the cortisol:cortisone ratio, a proxy measure for $11\beta$HSD activity, was lower in betamethasone-exposed placentae in both sexes.

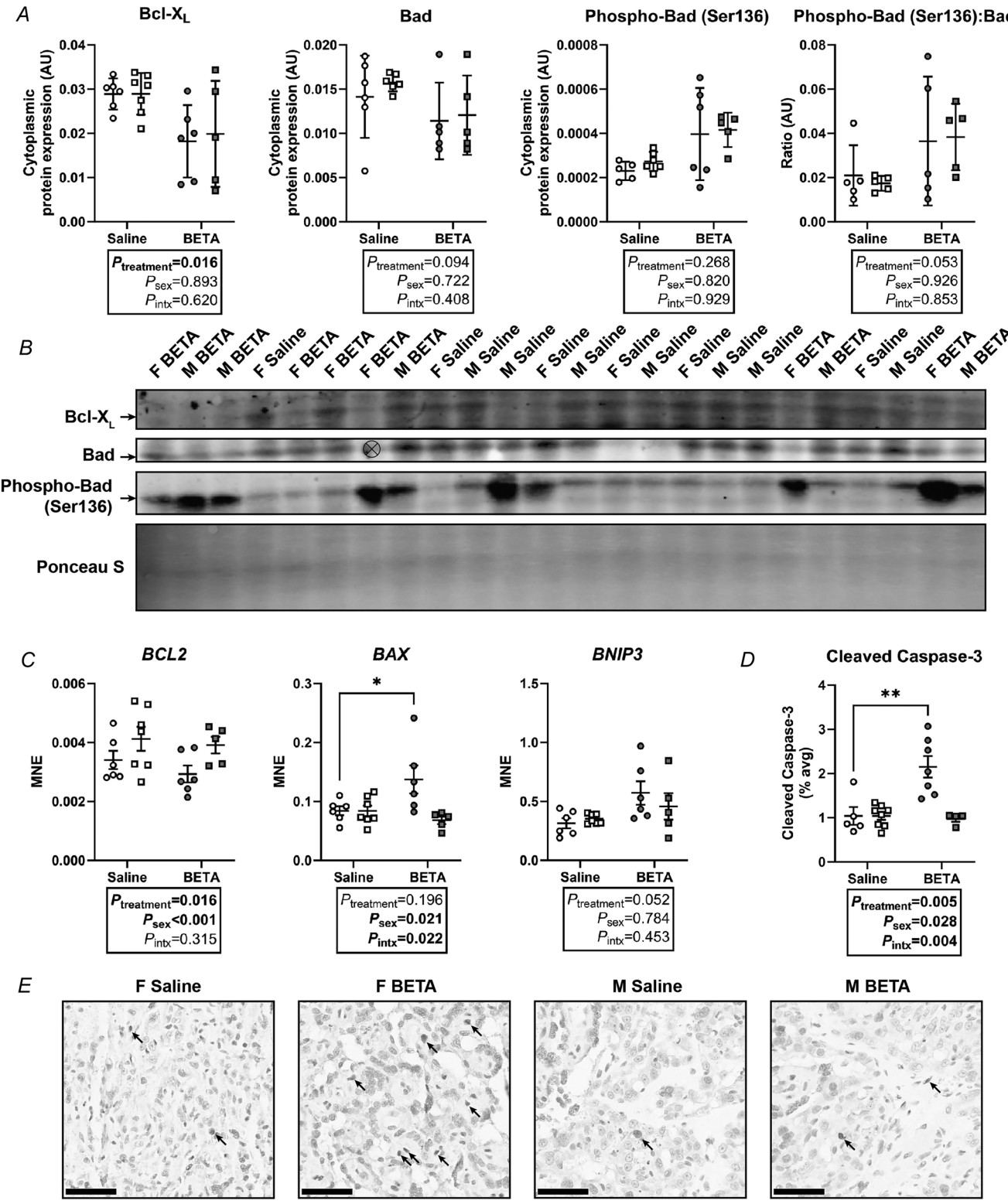

**Figure 5. Sex-specific induction of apoptotic pathways by betamethasone**
*A*, transcript expression of genes involved in apoptosis in saline [open symbols; female, *n* = 6 (circles), male, *n* = 7 (squares)] and betamethasone (BETA; closed symbols; female, *n* = 6, male, *n* = 5) placentae. *B* and *C*, cytoplasmic expression of Bcl-X$_L$, Bad, phospho-Bad (Ser136) and the ratio of phospho-Bad (Ser136):Bad in saline and BETA placentae. *D*, percentage of cleaved caspase-3 DAB$^+$ nuclei. *E*, representative micrographs of cleaved caspase-3-stained type B placentomes. Arrows indicate positive staining. Scale = 50 μm. Statistical analysis: linear mixed model (fixed factors: treatment and sex; random factor: ewe; intx = interaction effect) with Bonferroni

Collectively, our findings indicate that the placenta suppresses endogenous glucocorticoid signalling following betamethasone treatment via negative feed-back regulation of cortisol secretion and altered $11\beta$HSD activity.

Fetoplacental glucocorticoid signalling is not only modulated by the bioavailability of glucocorticoid ligands, but also by the differential expression and subcellular localisation of GR protein isoforms. Despite similar expression of *NR3C1*, the cytoplasmic expression of GR$\alpha$D, GR$\alpha$C and GR-P isoforms was higher following betamethasone treatment, but in female placentae only. The female-specific increase in GR$\alpha$C expression after betamethasone exposure is largely consistent with previous work by Cuffe et al. (2017), where maternal dexamethasone infusion from 12.5 to 14.5 dGA in rats led to higher nuclear GR$\alpha$C expression in female placentae only at 14.5 dGA. The expression of GR$\alpha$D isoforms

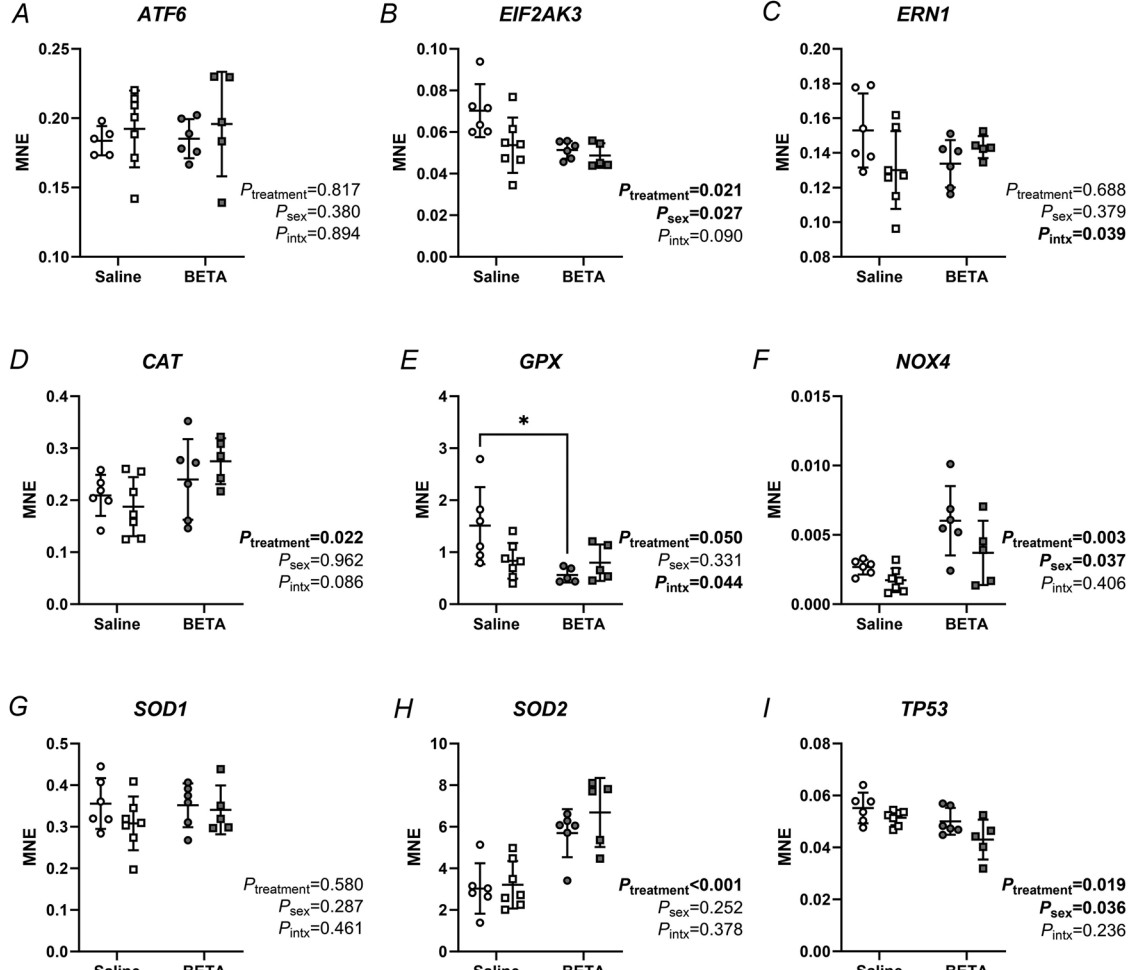

**Figure 6. Betamethasone and fetal sex impact markers of endoplasmic reticulum (ER) stress, DNA damage and response to hypoxia**

Transcript expression of (*A*) activating transcription factor 6 (*ATF6*), (*B*) eukaryotic translation initiation factor 2 alpha kinase 3 (*EIF2AK3*), (*C*) endoplasmic reticulum to nucleus signalling 1 (*ERN1*), (*D*) catalase (*CAT*), (*E*) glutathione peroxidase (*GPX*), (*F*) NADPH oxidase 4 (*NOX4*), (*G*) superoxide dismutase 1 (SOD1), (**H**) SOD2 and (*I*) *TP53* in female (circles) and male (squares) type B placentome samples from saline (open symbols, female, *n* = 6, male, *n* = 7) and betamethasone (BETA; closed symbols, female, *n* = 6, male, *n* = 5) treatment groups. Statistical analysis: linear mixed model (fixed factors: treatment and sex; random factor: ewe; intx = interaction effect) with Bonferroni *post hoc* test. *P* < 0.05 was considered statistically significant and is indicated in bold. **post hoc* *P* < 0.05. Symbols show data from individual animals, whereas bars and whiskers indicate the mean ± SD.

was also higher in guinea-pig placentae exposed to betamethasone than controls, but only in males (Saif et al., 2016). In humans, GRαC expression was higher in preterm compared to term placentae, regardless of sex or betamethasone treatment (Saif et al., 2015). GRαC enhances cellular sensitivity to glucocorticoids and mediates glucocorticoid-induced pro-apoptotic molecular signatures (Bender et al., 2013; Lu & Cidlowski, 2005), whereas GRαD has low transactivation capacity and is postulated to induce a state of glucocorticoid resistance (Lockett et al., 2024). Unlike GRαC and GRαD, which both function in a glucocorticoid-dependent manner to mediate transcriptional regulation of target genes, GR-P is unable to bind glucocorticoids and thus has a unique function: the enhancement of GRα function via GR isoform heterodimerisation (de Lange et al., 2010). However, whether GR-P enhances the function of GRα translational isoforms is not known. Collectively, our findings highlight a sexually dimorphic GR isoform response to betamethasone 48 h after administration in the sheep placenta, which may partly explain sex-specific molecular responses, such as apoptotic pathways, that are mediated by specific GR isoforms.

We also found sex-specific differences in placental apoptotic signalling pathways 48 h after betamethasone treatment. The enhanced female-specific pro-apoptotic response to betamethasone treatment is similar to that seen previously after dexamethasone exposure in mice (Cuffe et al., 2017). The lower expression of anti-apoptotic markers BCL-XL and *BCL2* (Xie et al., 2024) in betamethasone-exposed placentae was observed regardless of sex. We hypothesise that, when coupled with lower expression of *BCL2* in female than male placentae, and female-specific betamethasone-induced higher expression of apoptotic protein BAX, this probably results in betamethasone-induced apoptosis specifically in female placentae. A higher abundance of p53 in female than male placentae may further enhance apoptosis in female placentae (Chipuk et al., 2004). This apoptotic cascade culminates in activation of caspases and cellular apoptosis (Kasture et al., 2021), probably explaining the higher number of cleaved-caspase 3 positive nuclei in response to betamethasone in female placentae. Other markers of oxidative stress were similarly expressed more highly in females and induced by betamethasone in both sexes (*NOX4*) or suppressed by betamethasone in females only (*GPX*). Increased oxidative stress probably contributed to higher expression of the antioxidant enzyme catalase in betamethasone-exposed placentae (Schrader & Fahimi, 2006). Interestingly, catalase was upregulated by betamethasone-exposure in both sexes despite female-predominant effects of betamethasone on apoptotic markers. Although apoptosis was only 1–2% higher in betamethasone-exposed placentae, it should be noted that this was assessed only 48 h after treatment;

however, previous work in sheep that examined placental responses to early gestation ACS treatment (40–42 dGA) found dysregulated apoptotic signalling pathways at 100 dGA, and this was associated with reduced fetal growth in females only (Braun et al., 2015). Therefore, if ACS are administered and preterm delivery does not occur, but rates of apoptosis remain elevated, placental size and function may be impacted and contribute to the reduced placental and fetal growth often observed in human pregnancies after ACS treatment.

Although the present study highlights sex-specific responses to betamethasone treatment in the near-term sheep placenta, there are some recognised limitations. Most notably, the model used is not one of preterm birth but rather that of a mild phenotype of maternal allergic asthma. Because this study was not originally designed with placental development and function as the main outcome, treatment and tissue collection occurred near-term. Indeed, placental molecular and structural responses to preterm birth with or without ACS treatment are known (Couture et al., 2023; Paquette et al., 2023; Saif et al., 2016), but few studies have considered the sexual dimorphism of reported changes. In a rat model of experimental chorioamnionitis, a single dose of betamethasone prior to delivery improved placental function at the molecular and structural level, with similar responses to those reported in the present study (Netsanet et al., 2025); however, placental sex was not considered. This highlights the need for future studies to consider the independent and additive effects of preterm birth, betamethasone exposure and sex on placental molecular and structural functions. Likewise, asthma in pregnancy affects glucocorticoid signalling (Meakin et al., 2017) and may affect some of the placental responses reported herein; therefore, comparisons in the present study were between saline- and betamethasone-treated animals within the asthmatic group only. Another recognised limitation is the differences in sheep and human placental structure. Despite differences in gross phenotype, placental villus structure, gas exchange and uteroplacental oxygen consumption are similar in these two species (Flouri et al., 2022; Saini et al., 2021), as are impacts of ACS on fetal and placental growth (Elfayomy & Almasry, 2014; Moss et al., 2001). We acknowledge the importance of future studies that assess the independent and additive effects of preterm birth, betamethasone treatment and sex on placental molecular and structural responses at and beyond 48 h of treatment

## Conclusions

Using a clinically-relevant sheep model of near-term pregnancy, we demonstrated that betamethasone exposure for 48 h induces multiple changes to molecular

pathways in the placenta. In both sexes, betamethasone increased expression of pro-angiogenic factors and suppressed the expression of proliferative and growth markers. Of particular interest was the sexual dimorphism in expression of apoptotic regulators that suggest female placentae have impaired protection from betamethasone-induced apoptosis. The mechanism by which this sex-specific response to betamethasone exposure is mediated requires further investigation; however, the female-specific upregulation of the pro-apoptotic GR isoform, GR$\alpha$C, may partly explain these responses. Our findings of fetal sex-specific responses to betamethasone are consistent with those previously reported in humans and rodents, and suggest that conserved sex-specific adaptations to changes to the intrauterine environment may underpin growth and survival differences in the perinatal period.

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

## Additional information

### Data availability statement

The data generated and analysed during this study are available from the corresponding authors upon reasonable request.

### Competing interests

The authors declare that they have no competing interests.

### Author contributions

A.S.M., M.C.L., K.L.G. and J.L.M. were responsible for the conception or design of the work. A.S.M., M.C.L., S.L.H., J.L.R., V.L.C., C.T.R., M.D.W., K.L.G. and J.L.M. were responsible for acquisition or analysis or interpretation of data for the work. A.S.M., M.C.L., K.L.G. and J.L.M. were responsible for drafting the work or revising it critically for important intellectual content. All authors were responsible for approving the final version of the manuscript submitted for publication and agree to be accountable for all aspects of the work.

### Funding

The animal work was funded by a Channel 7 Children's Research Foundation Grant (20190628) to KLG, VLC and JLM. JLM and the molecular work were funded by an Australian Research Council Future Fellowship (Level 3; FT170100431). JLR was funded by an Australian Government Research Training Program Scholarships and Healthy Development Adelaide/Channel 7 Children's Research Foundation PhD Excellence Supplementary Scholarships. VLC was funded by a National Health and Medical Research Council (NHMRC) Senior Research Fellowship (APP1136100). CTR was funded by a NHMRC Investigator Grant (GNT1174971) and a Matthew Flinders Fellowship from Flinders University.

### Acknowledgements

We acknowledge the contribution of members of the Early Origins of Adult Health Research Group in assistance with tissue collection. We also acknowledge the technical assistance of the National Imaging Facility, an NCRIS capability, at PIRL, SAHMRI and their animal technicians.

Open access publishing facilitated by University of South Australia, as part of the Wiley - University of South Australia agreement via the Council of Australian University Librarians.

### Keywords

betamethasone, fetus, glucocorticoid receptor, placenta, sex differences, sheep

### Supporting information

Additional supporting information can be found online in the Supporting Information section at the end of the HTML view of the article. Supporting information files available:

**Peer Review History**

