## [Peer Review History · The Journal of Physiology]

Sex-specific effects of betamethasone on glucocorticoid and apoptotic signalling pathways in the sheep placenta

Ashley S Meakin, Mitchell C Lock, Stacey L Holman, Joshua L Robinson, Vicki L Clifton, Claire T. Roberts, Michael D Wiese, Kathryn L Gatford, and Janna L Morrison

DOI: [10.1113/JP289044](https://doi.org/10.1113/JP289044)

Corresponding author(s): Ashley Meakin (ashley.meakin@unisa.edu.au)

Review Timeline:

Submission Date:	09-Apr-2025
Editorial Decision:	13-May-2025
Revision Received:	27-Jun-2025
Accepted:	16-Jul-2025

Senior Editor: Laura Bennet

Reviewing Editor: Rebecca Simmons

Transaction Report:

Dear Dr Meakin,

Re: JP-RP-2025-289044 "Sex-specific effects of betamethasone on glucocorticoid and apoptotic signalling pathways in the sheep placenta" by Ashley S Meakin, Mitchell C Lock, Stacey L Holman, Joshua L Robinson, Vicki L Clifton, Claire T. Roberts, Michael D Wiese, Kathryn L Gatford, and Janna L Morrison

Thank you for submitting your manuscript to The Journal of Physiology. It has been assessed by a Reviewing Editor and by 2 expert referees and we are pleased to tell you that it is potentially acceptable for publication following satisfactory major revision.

REVISION CHECKLIST:

We look forward to receiving your revised submission.

Yours sincerely,

Laura Bennet
Senior Editor
The Journal of Physiology

REQUIRED ITEMS

- Author photo and profile. First or joint first authors are asked to provide a short biography (no more than 100 words for one author or 150 words in total for joint first authors) and a portrait photograph. These should be uploaded and clearly labelled together in a Word document with the revised version of the manuscript. See Information for Authors for further details.

- You must start the Methods section with a paragraph headed Ethical approval (https://jp.msubmit.net/cgi-bin/main.plex?form_type=display_requirements#methods).

Research must comply with The Journal's policies regarding animal experiments (<https://physoc.onlinelibrary.wiley.com/hub/animal-experiments>) and adherence to these policies must be stated in the manuscript.

Authors should confirm in their Methods section that their experiments were carried out according to the guidelines laid down by their institution's animal welfare committee, including an ethics approval reference number. The Methods section must contain a statement about access to food, water and housing, details of the anaesthetic regime: anaesthetic used, dose and route of administration, and method of killing the experimental animals.

EDITOR COMMENTS

Reviewing Editor:

There were several concerns raised by both reviewers including that clinically betamethasone is given for lung maturation in the setting of threatened preterm delivery and animals may deliver later than the 48hr time period selected here in animals at term. Responses seen in preterm placentas and with longer times of exposure may be very different than observations made here with short term exposure at term. Please address this in the Discussion.

Please also see 'Required Items' above.

REFEREE COMMENTS

Referee #1:

This paper describes a descriptive study of the changes in placental GR isoforms and cellular responses to betamethasone in pregnant sheep at term.

The choice of model is interesting, perhaps driven opportunistically by availability of tissue from another study. Why were

responses studied at term when in clinical practice betamethasone is usually employed in preterm scenarios to cause lung maturation in case of delivery? The placental response may be different at earlier gestational ages and also in vivo the placental responses seen may be occurring at later time points than the 48hrs studied here.

Introduction:

Lines 76-81. This description of placental function does not do justice to the many and complex roles of the placenta. As these authors are well aware and have stated the placenta is not just a conduit

Methods:

Line 150-155. This description appears to have been copied from another publication, all that is relevant here is collection of placental tissue.

Line 160. It is noted that only one section per placentome was scanned. What is the variability across several sections within a placentome?

Line 187, What species were the antibodies raised in? Rabbit?

Results:

Table 3 and Line 321-323, Arithmetic Barrier Thickness is derived from volume density of maternal epithelium divided by surface density of trophoblast. Could the combined error inherent in both measurements result in the finding of a significant difference in barrier thickness in female betamethasone exposed placentomes only?

Line 341. While it is stated II-6 and II-8 did not differ between treatments and sex II-6 between treatments was $p < 0.051$, perhaps this should be commented on.

Discussion:

The discussion is too long and could be edited down.

In discussing the relevance of the findings the authors could perhaps provide context and highlight that these observations are made immediately following 48hr of betamethasone treatment at term in contrast to the clinical scenario with treatment usually preterm and a longer (variable) period elapsing for effects on placental structure/function to arise (eg line 459). It is perhaps not surprising that minimal effects were seen on placental structure in this time frame (line 497). How might other responses studied be different when examined earlier in gestation when placental developmental stage is different?

Referee #2:

This study assesses the effect of betamethasone 48 h after administration on sheep placental molecular characteristics, under the hypothesis that it may be pro-angiogenic and there may be sex differences in the response to betamethasone. The study provides evidence that there are sex specific differences in pro-apoptotic markers in female placentas, with additional evidence in both sexes that there are increases in markers of angiogenesis and decreases in markers of growth and proliferation in response to this exposure. The article is well written and the methods/results align well with the interpretation of the data and are clearly put together.

First key point: "Betamethasone treatment for pregnancies at risk of preterm delivery reduces the risk of neonatal death, but off-target, sex-specific effects on placental function are known." - should be unknown?

Page 7, line 105/106 "In contrast, other studies report that ACS improve placental function.." Prior to this statement you have been specific about species, and it would be good to clarify species here.

Page 7 last paragraph. Hypothesis. As the evidence in the preceding paragraph is mixed, it is not entirely clear why the hypothesis would be that the response would be pro-angiogenic. Some clarifying statements would help.

Page 8, line 135. The methodologies state that the study used mildly asthmatic ewes, but this is not mentioned again that I can see. Revisiting this in the discussion around any implications that this has would provide context.

END OF COMMENTS

We thank the reviewing editor and reviewers for their comprehensive review of our work and have provided responses to all queries below. Line numbers provided for each change refer to the tracked changes version of the manuscript.

- Author photo and profile. First or joint first authors are asked to provide a short biography (no more than 100 words for one author or 150 words in total for joint first authors) and a portrait photograph. These should be uploaded and clearly labelled together in a Word document with the revised version of the manuscript. See Information for Authors for further details.

Thank you for bringing this to our attention. We have included a joint first author profile and portrait photographs in the resubmission.

- You must start the Methods section with a paragraph headed Ethical approval (https://jp.msubmit.net/cgi-bin/main.plex?form_type=display_requirements#methods).

Apologies for this oversight; the information has been shifted from the start of the animal model section to a separate section headed “Ethical approval”.

Research must comply with The Journal's policies regarding animal experiments (<https://physoc.onlinelibrary.wiley.com/hub/animal-experiments>) and adherence to these policies must be stated in the manuscript.

Thank you for your comment. This information was in the originally submitted manuscript, and in response to this comment we have shifted these statements regarding ethics to a separate section titled “Ethical approval”, at the start of the methods (see Lines 133-139).

Authors should confirm in their Methods section that their experiments were carried out according to the guidelines laid down by their institution's animal welfare committee, including an ethics approval reference number. The Methods section must contain a statement about access to food, water and housing, details of the anaesthetic regime: anaesthetic used, dose and route of administration, and method of killing the experimental animals.

We confirm that our experiments were carried out according to the South Australian Health and Medical Research Institute animal ethics committee (SAM455.19). Please refer to lines 133-139 for more details regarding ethical approval and compliance of this study.

We have added information regarding nutrition and animal housing to the manuscript as requested, please refer to lines 150-151.

Lines 150-151: *Animals were group-housed in outdoor paddocks with ad libitum access to water and natural pasture supplemented with oaten hay as required.*

Information on anaesthesia is included in the manuscript at lines 162-164, and the method of killing the animals is described on lines 166-167.

EDITOR COMMENTS

Reviewing Editor:

There were several concerns raised by both reviewers including that clinically betamethasone is given for lung maturation in the setting of threatened preterm delivery and animals may deliver later than the 48hr time period selected here in animals at term. Responses seen in preterm placentas and with longer times of exposure may be very different than observations made here with short term exposure at term. Please address this in the Discussion.

We agree that our study is not in the context of preterm birth and recognise that the acute molecular and structural responses reported in our study may be different to those observed in pregnancies treated with ACS that go on to deliver beyond 48 h after treatment. However, the guidelines for use of ACS recommend delivery of ACS 48 h prior to delivery and thus we selected this timepoint for study. We have reworked our final paragraph of the discussion to address these concerns, while noting that this point is also addressed throughout the discussion (lines 476-482, 524-527, 594-602). We conducted this study with the principles of the 3Rs in mind, which allowed us to understand the effect of betamethasone and sex on molecular and structural responses in the near-term sheep placenta. The final revised paragraph of the discussion stresses the importance of future studies that assess the independent and additive effects of preterm birth, betamethasone treatment, and sex on placental molecular and structural responses at and beyond 48 h of treatment.

Lines 607-615: *Indeed, placental molecular and structural responses to preterm birth with or without ACS treatment are known, but few studies have investigated whether these differ between sexes. In a rat model of experimental chorioamnionitis, a single dose of betamethasone prior to delivery improved placental function at the molecular and structural level, with similar responses to those reported in our current study (Netsanet et al., 2025); however, placental sex was not considered. This highlights a need for future studies to consider the independent and additive effects of preterm birth, betamethasone exposure, and sex on placental molecular and structural functions.*

Lines 626-634: *We acknowledge the importance of future studies that assess the independent and additive effects of preterm birth, betamethasone treatment, and sex on placental molecular and structural responses at and beyond 48 hours of treatment.*

Please also see 'Required Items' above.

The 'Required Items' have been addressed.

REFEREE COMMENTS

Referee #1:

This paper describes a descriptive study of the changes in placental GR isoforms and cellular responses to betamethasone in pregnant sheep at term.

The choice of model is interesting, perhaps driven opportunistically by availability of tissue from another study. Why were responses studied at term when in clinical practice betamethasone is usually employed in preterm scenarios to cause lung maturation in case of delivery? The placental response may be different at earlier gestational ages and also in vivo the placental responses seen may be occurring at later time points than the 48hrs studied here.

We thank the reviewer for their expert review and for raising this query. We conducted this study with the principles of the 3Rs in mind, which allowed us to understand the effect of betamethasone and sex on molecular and structural responses in the near-term sheep placenta, despite the primary outcome of the original study not being placenta-focussed. We agree that the placental responses may be different within a preterm scenario, where the pathophysiology that leads to preterm birth (e.g. infection and associated inflammatory response) may impact responses to betamethasone. We also agree that the impact of betamethasone on the placenta after 48 h may differ from that at longer periods. However, our study is near-term and appropriately models a proportion of human pregnancies that are administered antenatal corticosteroids later in gestation. Regardless, a unique aspect of our work is the ability to consider the effect of placental sex in the analysis.

We agree that the placental responses may be different at earlier gestational ages. Nevertheless, the pro-apoptotic responses to betamethasone in female placentae in our study are similar to previous work investigating placental molecular responses to early gestation dexamethasone treatment (40-42 dGA) at 50, 100, 125 and 140dGA. Braun *et al.* (2015) found subtle differences in markers of apoptosis (e.g. Caspase-3 mRNA) at 100 dGA that were indicative of dexamethasone-induced placental insufficiency and reduced fetal growth in females only. It would be interesting to study the independent and additive effects of preterm birth, sex, and betamethasone treatment on placental molecular and structural responses. Other studies have identified differences in the molecular profile in human placenta at term when compared with preterm (Couture *et al.*, 2023; Paquette *et al.*, 2023). Additionally, Saif *et al.* (2016) found that betamethasone treatment induced sex-specific differences in GR isoform profiles in guinea pig placenta collected at term when compared with preterm samples. However, no study has investigated these interactions in sheep.

We have reworked sections of our Discussion to address the abovementioned information.

Lines 596-599: "... however, previous work in sheep that examined placental responses to early gestation ACS treatment (40-42 dGA) found dysregulated apoptotic signalling pathways

at 100 dGA, and this was associated with reduced fetal growth in females only (Braun et al., 2015).”

Lines 607-610: Indeed, placental molecular and structural responses to preterm birth with or without ACS treatment are known (Saif et al., 2016; Couture et al., 2023; Paquette et al., 2023), but few studies have considered the sexual dimorphism of reported changes.

Introduction:

Lines 76-81. This description of placental function does not do justice to the many and complex roles of the placenta. As these authors are well aware and have stated the placenta is not just a conduit.

We agree and recognise that the role of the placenta extends beyond simply exchanging nutrients and waste between maternal and fetal circulations. In response to this suggestion, we have expanded the description of placental function to capture the complex nature of this organ in regulating fetal growth and development, as well as ensuring pregnancy success.

Lines 76-83: The placenta is a highly vascular, transient organ that is critical for maternal and fetal health, as well as pregnancy success. It not only supports fetal growth and development by exchanging nutrients and waste between maternal and fetal circulations, but also regulates the fetal and maternal endocrine and immune systems, protects the fetus from excess endogenous and exogenous chemical exposure, and is critical for the regulation of fetal programming and thus later life offspring health (Burton et al., 2016). Therefore, it is essential that normal placental function is maintained throughout pregnancy to ensure optimal fetal wellbeing.

Methods:

Line 150-155. This description appears to have been copied from another publication, all that is relevant here is collection of placental tissue.

We acknowledge that the description of Caesarean sections and tissue collection is similar to our previous work published from the same cohort of animals; however, details on anaesthesia and methods of human killing are required in the present manuscript as per journal requirements. We have revised this section and added information on placentome sampling as requested in a subsequent reviewer comment.

Lines 162-174: At 140 ± 2 dGA, fasted ewes were anaesthetised intravenously with ketamine (7 mL/kg) and diazepam (0.3 mL.kg⁻¹), intubated, and anaesthesia maintained with isoflurane (1.5-2.5% in air; Lyppards, SA, Australia). Fetuses were delivered by Caesarean section and ventilated for 45 min as previously reported (Robinson et al., 2024). Upon completion of lung function studies, ewes and lambs were humanely killed with sodium pentobarbitone (20 mg/kg, Virbac Australia, Peakhurst, NSW, Australia). Major maternal and neonatal organs were sampled as previously described (Clifton et al., 2016; McBride et al., 2021; Meakin et al., 2022) and stored for future studies, including phenotyping and sampling (Vatnick et al., 1991; Zhang et al., 2016) of placentomes from each pregnancy (saline: female

n=6, male n=8; betamethasone: female n=7, male n=5). This involved consistent sampling from the centre of each placentome to minimise variability between pregnancies. All morphometric and molecular analyses were performed on type B placentomes.

Line 160. It is noted that only one section per placentome was scanned. What is the variability across several sections within a placentome?

To the best of our knowledge, no study has examined the variability across several sections within a single sheep placentome. We acknowledge that other variables may affect placental morphometric measures – i.e., in humans, vascular density varies depending on location in relation to cord insertion (Aughwane *et al.*, 2019), and this may differ between tissue sections within a single biopsy. This is unlikely in ruminants, such as sheep and cows, where placental vascular density does not change depending on the distance from the umbilical cord (Reid *et al.*, 2022). In our study, all placentomes were Type B and processed using a standardised protocol at postmortem to collect tissue from a consistent section at the centre of each placentome, thereby minimising potential inter-animal variability. We also acknowledge that placental structure would vary depending on section plane (e.g. sagittal vs transverse). In the present study, all tissue samples were embedded in the same orientation and blocks were oriented to the sagittal plane prior to sectioning to further minimise inter-animal variability. We have amended our methodology to include this information.

Lines 172-173: *Cross-sections were consistently cut at the centre of each placentome to minimise variability between pregnancies.*

Line 177: *Sagittal sections (5 µm) from type B placentomes...*

Following staining; the entire cross-section of each placentome was digitised for analysis. Random-systematic sampled fields of view for counting were checked for potential processing or tissue damage before analysis. (see Figure 1 below for representative micrographs taken from different animals).

Figure 1. Representative micrographs of Type B placentomes from four separate animals demonstrating minimal variability of staining or structure between sections. Micrographs were taken at 20X magnification. Scale bar = 100µm.

Line 187, What species were the antibodies raised in? Rabbit?

We apologise that this information within the section describing the secondary antibody was difficult to find. The reviewer is correct that the primary antibody was raised in rabbits. This information can be derived from details on the secondary antibody used. Additionally, each primary antibody utilised in the study has its associated manufacturer code listed at first mention within the methods section.

Lines 219-220: “...with secondary antibody (Goat anti-Rabbit IgG (H+L) Secondary Antibody, Biotin, Thermo Fisher Scientific) and Streptavidin Protein, HRP (Thermo Fisher Scientific).”

Results:

Table 3 and Line 321-323, Arithmetic Barrier Thickness is derived from volume density of maternal epithelium divided by surface density of trophoblast. Could the combined error inherent in both measurements result in the finding of a significant difference in barrier thickness in female betamethasone exposed placentomes only?

We thank the reviewer for their comment. Though there is indeed inherent error in both measures, the observed standard deviation was not different between treatment groups or sexes (Figure 2, below). Barrier thickness measures using this formula have been utilised in previous publications (Roberts *et al.*, 2001; MacLaughlin *et al.*, 2005; Fletcher *et al.*, 2007; Zhang *et al.*, 2016; Clifton *et al.*, 2019) and this approach is recognised as standard practice for placental histology.

Figure 2. Visualisation of maternal epithelium and trophoblast volume density/placentome, and the barrier thickness of maternal epithelium in female (circles) and male (squares) type B placentomes from saline (open symbols, female n=6, male n=8) and betamethasone (BETA, closed symbols; female n=7, male n=4) treatment groups. The raw data for these graphs are presented within the manuscript within Table 3.

We have amended our ‘Placental morphometric measures’ section to include additional information regarding standard error.

Lines 186-188: The number of points necessary to achieve a standard error of less than 10 percent was calculated from a preliminary study using a nomogram relating test point number and volume density (Weibel *et al.*, 1966).

Line 341. While it is stated IL-6 and IL-8 did not differ between treatments and sex, IL-6 between treatments was $p < 0.051$, perhaps this should be commented on.

We acknowledge that the treatment P-value for IL-6 was 0.051, and that the sex P-values for both IL-6 and IL-8 were 0.830 and 0.325, respectively. Indeed, it may be that if additional samples were available, a significant reduction in IL-6 in response to betamethasone treatment could be observed. However, as stated in our *Statistical analysis* section of our *Methods* and as per Journal Policy, a $P < 0.05$ was considered statistically significant, and as per the journal policy, we have not speculated on trends.

Discussion:

The discussion is too long and could be edited down.

We thank the reviewer for this comment and have edited the discussion. In particular, we have condensed discussion on i) the angiogenic factors whose expression were impacted by betamethasone treatment (Paragraph 4), and ii) the negative feedback regulation of glucocorticoid release (Paragraph 6).

In discussing the relevance of the findings the authors could perhaps provide context and highlight that these observations are made immediately following 48hr of betamethasone treatment at term in contrast to the clinical scenario with treatment usually preterm and a longer (variable) period elapsing for effects on placental structure/function to arise (eg line 459 . It is perhaps not surprising that minimal effects were seen on placental structure in this time frame (line 497).

We agree that a limitation of this study is that it was performed in near-term sheep and that tissue was collected at 48 h after betamethasone treatment. Indeed, it would be interesting to study the independent and additive effects of preterm birth, sex, and betamethasone treatment on placental molecular and structural responses, particularly as other studies have identified changes in the molecular profile in human placenta at term when compared with preterm (Couture *et al.*, 2023; Paquette *et al.*, 2023). Additionally, Saif *et al.* (2016) found that betamethasone treatment induced sex-specific differences in GR isoform profiles in guinea pig placenta collected at term when compared with preterm samples. We have added this information when discussing the limitations of our study.

Lines 607-615: “*Indeed, placental molecular and structural responses to preterm birth with or without ACS treatment are known (Saif et al., 2016; Couture et al., 2023; Paquette et al., 2023), but few studies have considered the sexual dimorphism of reported changes. In a rat model of experimental chorioamnionitis, a single dose of betamethasone prior to delivery improved placental function at the molecular and structural level, with similar responses to those reported in our current study (Netsanet et al., 2025); however, placental sex was not considered. This highlights a need for future studies to consider the independent and additive effects of preterm birth, betamethasone exposure, and sex on placental molecular and structural functions.*”

How might other responses studied be different when examined earlier in gestation when placental developmental stage is different?

We thank the reviewer for raising this point. Although betamethasone was given near-term in the present study, there is some evidence that steroid administration induces similar changes even at earlier gestational ages. Work by Braun *et al.* (2015) examined placental molecular responses to early gestation dexamethasone treatment (40-42 dGA) at 50, 100, 125 and 140 dGA. The study found subtle differences in markers of apoptosis (e.g. Caspase-3 mRNA) at 100 dGA that were indicative of dexamethasone-induced apoptosis in female placentae only and reduced fetal growth. We have included this information in our revised discussion.

Lines 596-599: "... however, previous work in sheep that examined placental responses to early gestation ACS treatment (40-42 dGA) found dysregulated apoptotic signalling pathways at 100 dGA, and this was associated with reduced fetal growth in females only (Braun *et al.*, 2015)."

Referee #2:

This study assesses the effect of betamethasone 48 h after administration on sheep placental molecular characteristics, under the hypothesis that it may be pro-angiogenic and there may be sex differences in the response to betamethasone. The study provides evidence that there are sex specific differences in pro-apoptotic markers in female placentas, with additional evidence in both sexes that there are increases in markers of angiogenesis and decreases in markers of growth and proliferation in response to this exposure. The article is well written and the methods/results align well with the interpretation of the data and are clearly put together.

We thank the reviewer for their interest in our study.

First key point: "Betamethasone treatment for pregnancies at risk of preterm delivery reduces the risk of neonatal death, but off-target, sex-specific effects on placental function are known." - should be unknown?

We apologise for a lack of clarity and have reworded the key point accordingly:

Lines 33-35: *Betamethasone treatment for pregnancies at risk of preterm delivery reduces the risk of neonatal death, but also acts on glucocorticoid receptors (GR) in the placenta, inducing sex-specific changes that may impact function and fetal growth.*

Page 7, line 105/106 "In contrast, other studies report that ACS improve placental function." Prior to this statement you have been specific about species, and it would be good to clarify species here.

We thank the reviewer for bringing this to our attention and have amended this sentence to clarify the species:

Lines 109-112: *In contrast, other studies report that ACS improve placental function including decreased placental vascular resistance (Wallace & Baker, 1999) and accelerated*

placental villus maturation (Um-Bergström et al., 2018) in humans, and increased umbilical artery diameter and blood flow in mice (Cahill et al., 2019).

Page 7 last paragraph. Hypothesis. As the evidence in the preceding paragraph is mixed, it is not entirely clear why the hypothesis would be that the response would be pro-angiogenic. Some clarifying statements would help.

We thank the reviewer for raising this point and agree that our original hypothesis was not fully justified by the supporting information provided in the previous paragraph. We have reworded our hypothesis slightly to emphasise sex differences in placental responses to glucocorticoids are known and are likely conserved across species due to similarities in GR isoform profiles.

Lines 123-130: The expression profile of GR isoforms in the sheep placenta is similar to that of human placenta (Saif et al., 2014; Clifton et al., 2019). Because human female placentae are more glucocorticoid responsive than male placentae (Clifton, 2010; Saif et al., 2014), we hypothesised that betamethasone exposure would induce sex-specific differences to molecular and structural measures in the near-term sheep placenta, with a greater response observed in females than males.

Page 8, line 135. The methodologies state that the study used mildly asthmatic ewes, but this is not mentioned again that I can see. Revisiting this in the discussion around any implications that this has would provide context.

We thank the reviewer for raising this point. Our team is aware that moderate-severe asthma during human pregnancy can impact placental function in a sex-specific manner. Unlike previous studies from our team that examined placental molecular responses to a moderate sheep model of maternal asthma (Clifton *et al.*, 2019), samples derived in the current study are from a model of mild allergic asthma that did not result in an increase in maternal eosinophil count or abundance of pro-inflammatory cytokines in the circulation, but were treated with either saline or betamethasone. In response to this comment, we have addressed the use of this model in the revised Discussion.

Lines 615-618: Likewise, asthma in pregnancy affects glucocorticoid signalling (Meakin et al., 2017) and may affect some of the placental responses reported herein; therefore, comparisons in the current study were between saline and betamethasone treated animals within the asthmatic group only.

References

- Aughwane R, Schaaf C, Hutchinson JC, Virasami A, Zuluaga MA, Sebire N, Arthurs OJ, Vercauteren T, Ourselin S, Melbourne A & David AL. (2019). Micro-CT and histological investigation of the spatial pattern of feto-placental vascular density. *Placenta* **88**, 36-43.
- Braun T, Meng W, Shang H, Li S, Sloboda DM, Ehrlich L, Lange K, Xu H, Henrich W, Dudenhausen JW, Plagemann A, Newnham JP & Challis JR. (2015). Early dexamethasone treatment induces placental apoptosis in sheep. *Reproductive sciences (Thousand Oaks, Calif)* **22**, 47-59.
- Cahill LS, Whitehead CL, Hobson SR, Stortz G, Kingdom JC, Baschat A, Murphy KE, Serghides L, Macgowan CK & Sled JG. (2019). Effect of maternal betamethasone administration on feto-placental vascular resistance in the mouse†. *Biol Reprod* **101**, 823-831.
- Clifton VL, McDonald M, Morrison JL, Holman SL, Lock MC, Saif Z, Meakin A, Wooldridge AL, Gatford KL, Wallace MJ, Muhlhausler BS, Bischof RJ & Moss TJM. (2019). Placental glucocorticoid receptor isoforms in a sheep model of maternal allergic asthma. *Placenta* **83**, 33-36.
- Couture C, Brien M-E, Boufaied I, Duval C, Soglio DD, Enninga EAL, Cox B & Girard S. (2023). Proinflammatory changes in the maternal circulation, maternal–fetal interface, and placental transcriptome in preterm birth. *American Journal of Obstetrics and Gynecology* **228**, 332.e331-332.e317.
- Fletcher CJ, Roberts CT, Hartwich KM, Walker SK & McMillen IC. (2007). Somatic cell nuclear transfer in the sheep induces placental defects that likely precede fetal demise. *Reproduction* **133**, 243-255.
- MacLaughlin SM, Walker SK, Roberts CT, Kleemann DO & McMillen IC. (2005). Periconceptual nutrition and the relationship between maternal body weight changes in the periconceptual period and feto-placental growth in the sheep. *The Journal of Physiology* **565**, 111-124.
- Paquette AG, MacDonald J, Bammler T, Day DB, Loftus CT, Buth E, Mason WA, Bush NR, Lewinn KZ, Marsit C, Litch JA, Gravett M, Enquobahrie DA & Sathyanarayana S. (2023). Placental transcriptomic signatures of spontaneous preterm birth. *Am J Obstet Gynecol* **228**, 73.e71-73.e18.

- Reid DS, Burnett DD, Contreras-Correa ZE & Lemley CO. (2022). Differences in bovine placentome blood vessel density and transcriptomics in a mid to late-gestating maternal nutrient restriction model. *Placenta* **117**, 122-130.
- Roberts CT, Sohlstrom A, Kind KL, Earl RA, Khong TY, Robinson JS, Owens PC & Owens JA. (2001). Maternal Food Restriction Reduces the Exchange Surface Area and Increases the Barrier Thickness of the Placenta in the Guinea-pig. *Placenta* **22**, 177-185.
- Saif Z, Dyson RM, Palliser HK, Wright IM, Lu N & Clifton VL. (2016). Identification of Eight Different Isoforms of the Glucocorticoid Receptor in Guinea Pig Placenta: Relationship to Preterm Delivery, Sex and Betamethasone Exposure. *PloS one* **11**, e0148226.
- Um-Bergström M, Papadogiannakis N, Westgren M & Vinnars M-T. (2018). Antenatal corticosteroid treatment and placental pathology, with a focus on villous maturation. *Acta obstetricia et gynecologica Scandinavica* **97**, 74-81.
- Wallace EM & Baker LS. (1999). Effect of antenatal betamethasone administration on placental vascular resistance. *The Lancet* **353**, 1404-1407.
- Zhang S, Barker P, Botting KJ, Roberts CT, McMillan CM, McMillen IC & Morrison JL. (2016). Early restriction of placental growth results in placental structural and gene expression changes in late gestation independent of fetal hypoxemia. *Physiol Rep* **4**, e13049.

Dear Dr Meakin,

Re: JP-RP-2025-289044R1 "Sex-specific effects of betamethasone on glucocorticoid and apoptotic signalling pathways in the sheep placenta" by Ashley S Meakin, Mitchell C Lock, Stacey L Holman, Joshua L Robinson, Vicki L Clifton, Claire T. Roberts, Michael D Wiese, Kathryn L Gatford, and Janna L Morrison

We are pleased to tell you that your paper has been accepted for publication in The Journal of Physiology.

Yours sincerely,

Laura Bennet
Senior Editor
The Journal of Physiology

If you would like to receive our 'Research Roundup', a monthly newsletter highlighting the cutting-edge research published in The Physiological Society's family of journals (The Journal of Physiology, Experimental Physiology, Physiological Reports, The Journal of Nutritional Physiology and The Journal of Precision Medicine: Health and Disease), please click this link, fill in your name and email address and select 'Research Roundup':
<https://www.physoc.org/journals-and-media/membernews>

- You can help your research get the attention it deserves! Check out Wiley's free Promotion Guide for best-practice recommendations for promoting your work at: www.wileyauthors.com/eo/guide. You can learn more about Wiley Editing Services which offers professional video, design, and writing services to create shareable video abstracts, infographics, conference posters, lay summaries, and research news stories for your research at: www.wileyauthors.com/eo/promotion.

EDITOR COMMENTS

Reviewing Editor:

Thank you for submitting your manuscript to the Journal of Physiology. We appreciate that you addressed the reviewers concerns.

REFEREE COMMENTS

Referee #1:

The authors have revised their manuscript and tempered the discussion in view of the previous critique. They appropriately highlight the strengths and acknowledge the limitations of the work.

Referee #2:

I thank the authors for their detailed responses to my comments, which they have addressed to my satisfaction and I have no further comments.